# Dynamic corticothalamic modulation of the somatosensory thalamocortical circuit during wakefulness

Elaida D. Dimwamwa [1], Aurélie Pala [1,2], Vivek Chundru [1], Nathaniel C. Wright [1] & Garrett B. Stanley [1] ✉

The feedback projections from cortical layer 6 (L6CT) to the sensory thalamus have long been implicated in playing a primary role in gating sensory signaling but remain poorly understood. To causally elucidate the full range of effects of these projections, we targeted silicon probe recordings to the whisker thalamocortical circuit of awake mice selectively expressing Channelrhodopsin-2 in L6CT neurons. Through optogenetic manipulation of L6CT neurons, multi-site electrophysiological recordings, and modeling of L6CT circuitry, we establish L6CT neurons as dynamic modulators of ongoing spiking in the ventral posteromedial nucleus of the thalamus (VPm), either suppressing or enhancing VPm spiking depending on L6CT neurons' firing rate and synchrony. Differential effects across the cortical excitatory and inhibitory sub-populations point to an overall influence of L6CT feedback on cortical excitability that could have profound implications for regulating sensory signaling across a range of ethologically relevant conditions.

Amid the traditionally studied feedforward neuronal pathways underlying sensory perception are numerous feedback processes that are much less well understood. Layer 6 corticothalamic (L6CT) feedback neurons of all primary sensory cortices except the olfactory cortex are one such feedback process. These highly convergent, glutamatergic cortical neurons contribute ~30-50% of the synapses in sensory first-order (FO) thalamic nuclei[1–4]. Thus, L6CT neurons provide a potent modulatory input on thalamic excitability that likely plays a critical yet enigmatic role in thalamocortical sensory signaling.

In addition to providing direct, monosynaptic excitation to the FO thalamus, L6CT neurons also provide disynaptic inhibition via axon collaterals targeted to the inhibitory thalamic reticular nucleus (TRN), which, in turn, projects to the FO thalamus. Studies in thalamocortical brain slices have shown that the L6CT-mediated confluence of monosynaptic excitation and disynaptic inhibition in the FO thalamus enables a rich set of potential effects on thalamic excitability[5,6]. While several recent in-vivo studies have attributed L6CT neurons to either enhancing *or* suppressing the gain of FO thalamic sensory responses[7–14], the full dynamic range of L6CT modulation on FO

thalamus is poorly understood during wakefulness[9,15]. Moreover, cortical neurons integrate the modulated thalamic inputs as well as direct modulatory effects from intracortical L6CT projections. The effect of the full range of L6CT modulation on cortical neuron activity during wakefulness remains unknown (but see Olsen et al.[11]).

Starting as early as the pioneering neuroanatomical work of Ramón y Cajal[16], a plethora of studies have outlined the intricate anatomical details of the L6CT circuit[17–20]. However, it is only from recent work that we are beginning to understand the precise contexts in which these neurons become engaged and exert their influence, expanding the range of observed L6CT activity patterns. In addition to being responsive to sensory stimuli[21–25], L6CT neuron activity is also modulated by extra-sensory features, such as head rotation[26], motor preparation[25], whisking[27], and locomotion[21]. Additionally, L6CT neurons serve as a circuit mechanism for brain state-dependent modulation of the FO thalamus[28]. Collectively, these findings point to a continuous and context-dependent feedback modulation of the FO thalamus that acts through the regulation of L6CT population activity across a range of behaviors. Yet a unified understanding of the direct

[1]Wallace H Coulter Department of Biomedical Engineering, Georgia Institute of Technology and Emory University, Atlanta, GA, USA. [2]Department of Biology, Emory University, Atlanta, GA, USA. ✉e-mail: garrett.stanley@bme.gatech.edu

effects of this broadly acting regulation mechanism on thalamocortical structures remains elusive.

To causally uncover the full range of influence that L6CT neurons have on ongoing and stimulus-evoked spiking activity throughout the sensory thalamocortical circuit during wakefulness, we targeted silicon probe recordings to the whisker region of the primary

**Awake, NTSR1-Cre mouse with rAAV5-DIO-hCHR2-EYFP**

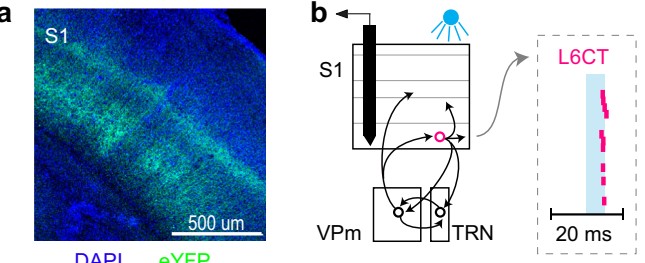

**LED activation of an example L6CT neuron**

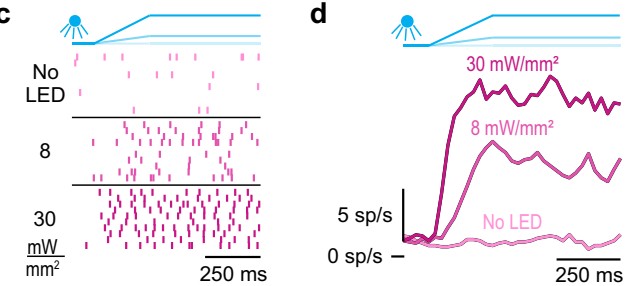

**LED activation of the L6CT population (61 neurons)**

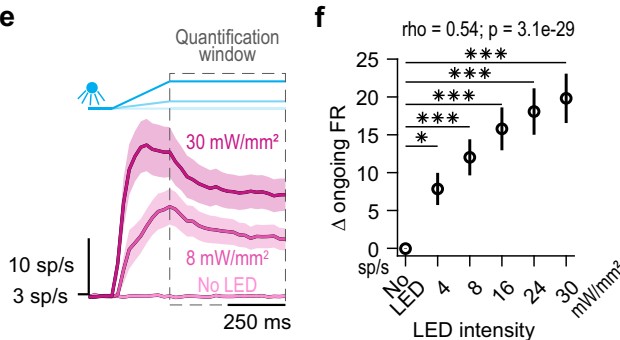

**Fig. 1 | Optogenetic activation of L6CT neurons in awake mice. a** A coronal section of DAPI labelled S1 neurons (blue) and selective ChR2-eYFP expression in L6CT neuron processes (green) of NTSR1-Cre mice (expression verified in all mice; $n = 11$ mice). **b** The direct effect of the LED on L6CT neuron activity is measured from in vivo, silicon probe recordings of neuronal spiking in S1 of awake, head-fixed, NTSR1-Cre mice ($n = 36$ recordings). L6CT neurons are activated by an optical fiber positioned at the cortical surface and are identified by their low latency and low jitter spiking response to 5 ms LED pulses (right inset raster; see Methods). **c** Raster plots of an example L6CT neuron activated by 0, 8, and 30 mW/mm² LED inputs. This neuron's spiking increases to 8 and 30 mW/mm² LED inputs. **d** PSTHs of the same L6CT neuron in (**c**). **e** Grand PSTHs of the L6CT population to 0, 8, and 30 mW/mm² LED inputs (mean +/− sem across neurons). The population's firing rate is enhanced by increasing LED intensity. **f** The mean +/− sem change in the ongoing firing rate of the L6CT population contributing to the PSTHs in **e** is monotonically enhanced by increasing LED intensities ($n = 61$ neurons; 4 mW/mm²: $p = 9.8e-3$; 8 mW/mm² $p = 6.9e-6$; 16 mW/mm²: $p = 1.8e-10$; 24 mW/mm² $p = 9.7e-12$; 30 mW/mm² $p = 9.7e-12$; $p$ value indicates pairwise comparison to No LED condition using the two-sided Wilcoxon signed rank test with a Bonferroni correction; * indicates $p < 1.0e-2$; *** indicates $p < 2.0e-4$; rho & $p$-value: Spearman's rank correlation).

somatosensory cortex (S1), the ventral posteromedial nucleus (VPm), and the thalamic reticular nucleus (TRN) of awake, NTSR1-cre mice selectively expressing Channelrhodopsin-2 (ChR2) in L6CT neurons. We found that increasing optogenetic activation of L6CT neurons shifted from suppressing to enhancing VPm ongoing activity, which we refer to as bidirectional modulation of VPm. The findings here suggest that L6CT-mediated bidirectional modulation of VPm activity is likely a combination of indirect contributions from L6CT activation of TRN and direct contributions to VPm through changes in L6CT population firing rate as well as spiking synchrony. Differential effects of L6CT activation on ongoing regular-spiking (RS) and fast-spiking (FS) cortical neurons' activity result in an increase in overall cortical excitability. Further, we found that L6CT neuron activation enhances the magnitude of the VPm sensory response and leaves the sensory response magnitude of the RS S1 population unchanged. Yet, the sensory response magnitude of the FS S1 population is maximally enhanced at the lower but not higher optogenetic drive of L6CT neurons. These effects are reminiscent of the dynamic changes observed in the ongoing activity of VPm neurons that would then be inherited by S1, resulting in the differential L6CT engagement of cortical excitatory and inhibitory subpopulations. These results highlight L6CT neurons as rate and synchrony-dependent, dynamic modulators of thalamic and cortical excitability.

## Results

To investigate the direct modulatory effect of L6CT neurons on thalamocortical spiking activity during wakefulness, we injected a recombinant adeno-associated virus (rAAV) encoding Cre-dependent ChR2 fused with eYFP into the whisker region of S1 of NTSR1-Cre mice[29] (Fig. 1a). ChR2-expressing L6CT neurons were optically driven with ramp-and-hold LED inputs (250 ms ramp, 500 ms hold durations) across a range of intensities using a 470 nm LED coupled to a 400 um diameter optical fiber positioned at the S1 surface. Concurrently, we targeted silicon probe recordings to S1 and measured the spiking activity of populations of individual L6CT neurons. L6CT neurons were distinguished from other S1 neurons based on a statistical measure of their short latency and low jitter spike activation to a 5 ms, square LED pulse, quantified using the stimulus-associated spike latency (SALT) test[30] (Fig. 1b). The analyses of neuronal activity both here and in subsequent analyses are during periods of whisker quiescence as determined from videography (see Methods).

For the example L6CT neuron presented in Fig. 1c, increasing LED intensity increases the neuron's spiking. A relatively rapid rise in firing rate during the LED ramp followed by steady-state spiking during the hold period of the LED input is apparent in both the individual neuron population peri-stimulus time histograms (PSTHs; Fig. 1d) as well as the population grand PSTH (Fig. 1e). Here and in subsequent analyses, we quantified the effects of the LED activation during this period of steady-state activity of the L6CT population, as highlighted by the quantification window in Fig. 1e. Across the population of recorded L6CT neurons, the measured change in ongoing firing rate increases monotonically with LED intensity (Fig. 1f). Note that the change is relative to a fairly low baseline level of firing of approximately 2.6 spikes/second of the L6CT neurons (compared to a baseline level of firing of 4.9 spikes/second for the rest of the S1 RS neuron population). Also note that the elicited changes in spiking activity were not due to the LED illumination alone, as verified in control experiments conducted in a wild-type mouse (Figure S1).

### Increasing optogenetic activation of L6CT neurons results in bidirectional modulation of ongoing firing rates in the VPm

We next evaluated the direct effect of L6CT neuron activation on the FO thalamus by targeting silicon probe recordings to the ventral posteromedial (VPm) nucleus of the thalamus and measuring the spiking activity of populations of individual VPm neurons during

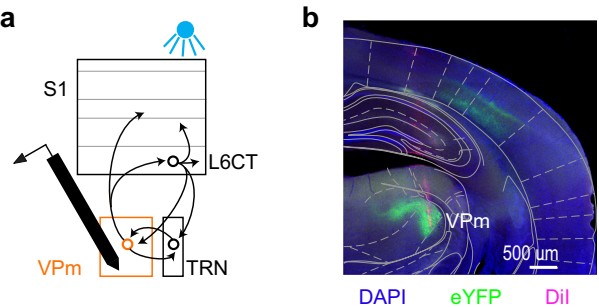

**a**

**b**

**LED modulation of an example VPm neuron**

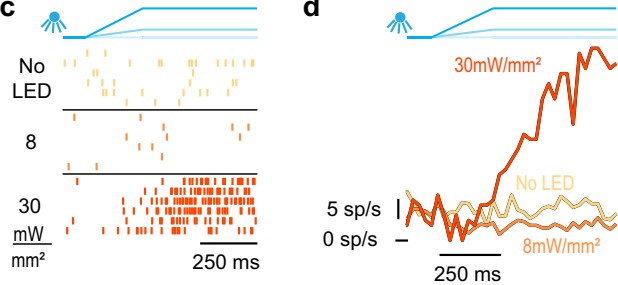

**c**  **d**

**LED modulation of the VPm population (75 neurons)**

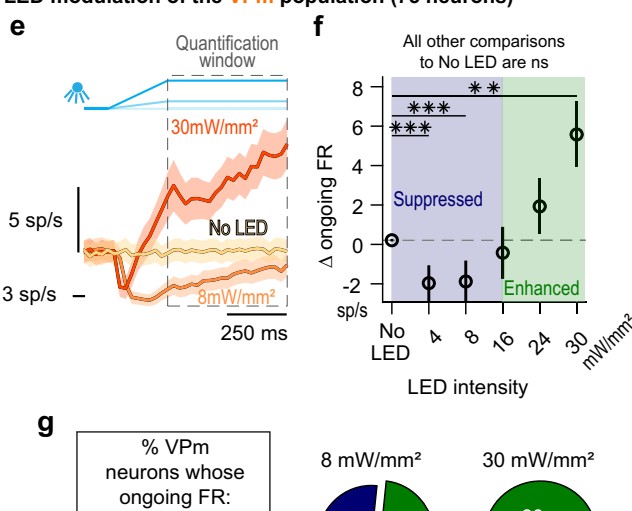

**e**  **f**

**g**

**Fig. 2 | Increasing optogenetic activation of L6CT neurons results in bidirectional modulation of ongoing firing rates in the VPm. a** The direct effect of L6CT neuron activation on VPm is measured from in vivo, silicon probe recordings of neuronal spiking in the VPm of awake NTSR1-Cre mice selectively expressing ChR2 in L6CT neurons ($n = 26$ recordings). **b** A coronal section of DAPI labelled neurons (blue), ChR2-eYFP expression in L6CT axons terminating in the VPm (green), and a VPm recording tract labelled with DiI (magenta; Allen Brain Atlas: −2.06 mm from Bregma; expression verified in all mice; $n = 13$ mice). **c** Raster plots of an example VPm neuron modulated by optogenetic activation of L6CT neurons at 0, 8, and 30 mW/mm². This neuron's spiking decreases to 8 mW/mm² but increases to 30 mW/mm² optogenetic activation of L6CT neurons. **d** PSTHs of the same VPm neuron in (**c**), indicating that this neuron's firing rate is bidirectionally modulated by increasing optogenetic activation of L6CT neurons: its firing rate is suppressed relative to the baseline to 8 mW/mm², but is enhanced relative to the baseline to 30 mW/mm² activation of L6CT neurons. **e** Grand PSTHs of the VPm population modulated by optogenetic activation of L6CT neurons at 0, 8, and 30 mW/mm² (mean +/− sem across neurons). The population's ongoing activity is bidirectionally modulated by increasing optogenetic activation of L6CT neurons. **f** The mean +/− sem change in ongoing firing rate of the VPm population contributing to the PSTHs in **e** is bidirectionally modulated in a graded manner across LED intensities, with a reversal in the trend occurring beyond 8 mW/mm² ($n = 75$ neurons; 4 mW/mm²: $p = 8.7e{-}5$; 8 mW/mm²: $p = 1.5e{-}4$; 30 mW/mm²: $p = 1.3e{-}3$; 16 mW/mm²: $p = 4.3e{-}2$; 24 mW/mm²: $p = 2.6e{-}1$; 30 mW/mm²: $p = 1.3e{-}3$; $p$ value indicates pairwise comparisons to No LED condition using the two-sided Wilcoxon signed rank test with a Bonferroni correction; ** indicates $p < 2e{-}3$; *** indicates $p < 2e{-}4$). **g** Counting the modulatory effect on individual VPm neurons indicates that a majority of neurons are suppressed at 8 mW/mm², whereas a majority of neurons are enhanced at 30 mW/mm² (Change for each neuron was determined by the two-sided Wilcoxon signed rank test on pre- and post-LED input firing rates across trials).

30 mW/mm² (Fig. 2e). We first characterized the overall effect of the L6CT activation on the VPm by quantifying the population spiking activity during the same quantification window used on L6CT neurons, which is during the hold period of the LED input. Across the five presented LED intensities, the change in the ongoing firing rate of VPm neurons is bidirectionally modulated in a graded manner, with relatively low LED intensities suppressing the VPm population firing rate, and a reversal of this suppressive effect occurring at 16 mW/mm², beyond which higher LED intensities enhance the firing rate of the VPm population (Fig. 2f).

The data presented here only includes trials with no whisker movement, a factor that is well known to be associated with changes in global brain state and modulation of VPm spiking[36]. Analysis of trials with whisker movement shows that the effect of the LED activation of L6CT neurons on the spontaneous activity of VPm neurons is nearly identical to that for trials without whisker movement (Figure S3).

When counting individual neurons, the ongoing firing rate of 60% of recorded neurons (45 of 75) is suppressed at 8 mW//mm², whereas 60% (45 of 75) of the same set of recorded neurons are enhanced at 30 mW/mm² (Fig. 2g). This heterogeneity of effects can be observed in simultaneously recorded VPm neurons with preferential responsiveness to the same whisker. Further, bidirectional modulation can be observed in simultaneously recorded neurons with preferential responsiveness to differing whiskers (Figure S4). Thus, likely due to the broad spatial nature of our LED input, the population heterogeneity is not trivially explained by the functional topography of L6CT axonal projections.

Despite L6CT neurons being in a steady-state during the hold period of the LED inputs, spiking in the VPm neuron population is not. This is apparent in Fig. 2e where the firing rate in the 8 mW/mm² condition increases slightly during the hold period of the LED input. However, even when restricting the analysis to the last 50 ms of the LED input, the VPm population is still significantly suppressed by the 8 mW/mm² LED input.

optogenetic activation of L6CT neurons with increasing LED intensity (Fig. 2a). The identification of VPm neurons recorded at the relevant stereotaxic coordinate was validated by histological verification and measures of their sensory response and spike waveform width[31–35] (Fig. 2b & S2; see Methods).

For the example VPm neuron presented in Fig. 2c, d, 8 mW/mm² optogenetic activation of L6CT neurons results in spiking activity that is persistently below the baseline VPm rate of approximately 6.3 spikes/second. However, at 30 mW/mm², the same VPm neuron's spiking activity transiently decreases during the ramping portion of the LED input, before increasing and staying above baseline levels for the remainder of the LED input.

Across the population, we found that optogenetic activation of L6CT neurons bidirectionally modulates the ongoing firing rate of VPm neurons. That is, overall, VPm neurons are suppressed relative to baseline at 8 mW/mm² but are enhanced relative to baseline at

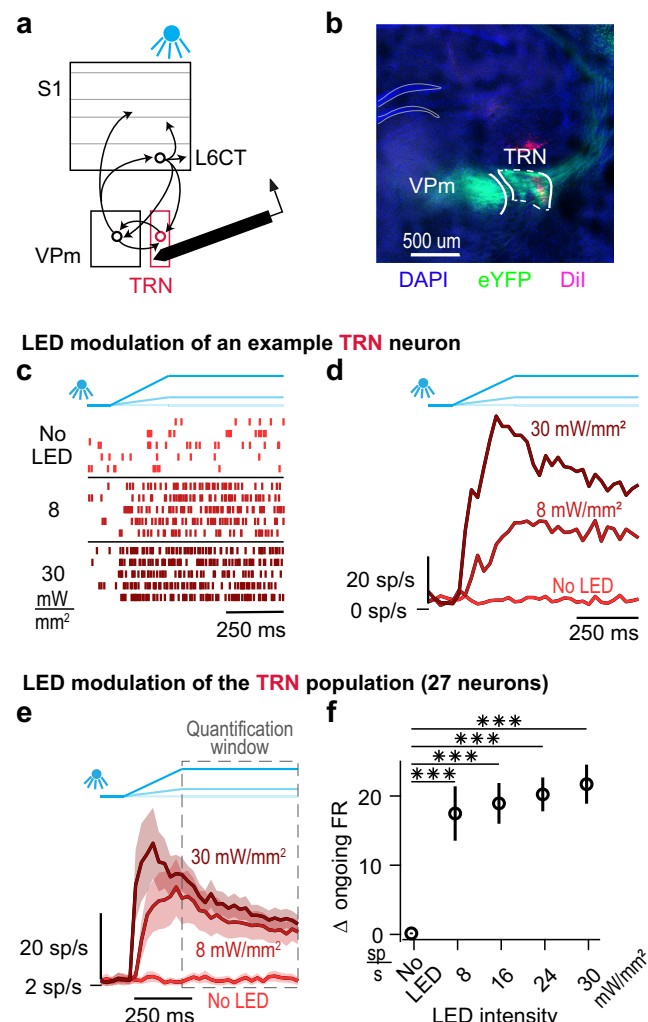

**LED modulation of an example TRN neuron**

**LED modulation of the TRN population (27 neurons)**

**Fig. 3 | Optogenetic activation of L6CT neurons enhances ongoing firing rates in the TRN. a** The direct effect of L6CT neuron activation on TRN is measured from in vivo, silicon probe recordings of neuronal spiking in the somatosensory section of TRN of awake NTSR1-Cre mice selectively expressing ChR2 in L6CT neurons (*n* = 9 recordings). **b** A coronal section of DAPI labelled neurons (blue), ChR2-eYFP expression in L6CT axons terminating in the VPm and TRN (green), and a TRN recording tract labelled with DiI (magenta; Allen Brain Atlas: −1.7 mm from Bregma; expression verified in all mice; *n* = 6 mice). **c** Raster plots of an example TRN neuron modulated by optogenetic activation of L6CT neurons at 0, 8, and 30 mW/mm². This neuron's spiking increases to 8 and 30 mW/mm² optogenetic activation of L6CT neurons. **d** PSTHs of the same TRN neuron in (**c**). **e** Grand PSTHs of the TRN population modulated by optogenetic activation of L6CT neurons at 0, 8, and 30 mW/mm² (mean +/− sem across neurons). The population's firing rate is strongly enhanced by both LED intensities. **f** The mean +/− sem change in ongoing firing rate of the TRN population contributing to the PSTHs in e is increased across all LED intensities (*n* = 27 neurons; 4 mW/mm²: *p* = 5.6e−6; 8 mW/mm²: *p* = 5.6e−6; 16 mW/mm²: *p* = 5.6e−6; 24 mW/mm²: *p* = 5.6e−6; 30 mW/mm²: *p* = 5.6e−6; *p* value indicates pairwise comparisons to no LED condition using the two-sided Wilcoxon signed rank test with a Bonferroni correction; *** indicates *p* < 2.5e−4).

## Optogenetic activation of L6CT neurons enhances ongoing firing rates in the TRN

Given that the thalamic reticular nucleus (TRN) is the primary source of L6CT-mediated inhibition in the VPm[5,6,37,38], it is important to consider the potential role of TRN in our observations. The rate of L6CT-mediated enhancement of TRN spiking relative to the rate of L6CT activation could play a role in the bidirectional modulation of VPm. To test this, we turned to silicon probe recordings of the spiking activity of populations of individual TRN neurons during optogenetic

activation of L6CT neurons (Fig. 3a). The identification of TRN neurons recorded at the relevant stereotaxic coordinate was validated by histological verification as well as their spike waveform width and measures of their sensory response and modulation from LED activation of L6CT neurons[9,39–43] (Fig. 3b & S2).

The spiking of the example TRN neuron presented in Fig. 3c, d is strongly increased by the LED at both 8 mW/mm² and 30 mW/mm². This strong effect of the LED modulation across all LED intensities largely holds for all recorded TRN neurons (Figure S5). Again, quantifying the population effect during steady-state L6CT activation, or the quantification window, we observe an enhancement of TRN spiking across all LED intensities. Further, among the LED-on conditions, increasing LED intensity causes relatively small increases in TRN firing rates (Fig. 3e, f). Relative to the monotonic increase in L6CT firing rate with LED intensity that likely induces a similarly increasing excitatory conductance in VPm, the inhibitory conductance driven through TRN likely exhibits a more modest dependence on L6CT activation. This coupled with the strong activation of TRN at the lowest LED intensity likely contributes to the bidirectional influence of L6CT on VPm (see Discussion).

## Optogenetic inputs of increasing LED intensity monotonically enhance the synchrony of the L6CT population

Neurons in circuits with inherently delayed, disynaptic inhibition and highly convergent excitatory synaptic inputs are extremely sensitive to the coordinated timing of those inputs, or the synchrony, making this element of population activity perhaps as important as overall firing rate[35,44–48]. Given the disynaptic inhibition in the L6CT circuit and to further resolve the L6CT-mediated bidirectional changes in VPm ongoing spiking activity, we assessed the synchrony of the L6CT population during our optogenetic manipulations (Fig. 4a).

Synchronous L6CT spikes are defined as those that occur within 7.5 ms of a spike from another simultaneously recorded L6CT neuron. To quantify synchrony, since firing rate and synchrony covary, we measured the synchrony strength or the rate of synchronous spikes normalized by the overall firing rate of the pair of neurons[34,49] (Fig. 4b; see Methods). The synchrony strength of the L6CT population is also monotonically enhanced by increasing LED intensity. This effect is exhibited in the grand PSTH of the L6CT population synchrony, with a rapid, transient increase in synchrony strength during the optogenetic ramp that is sustained during the hold period (Fig. 4c). The synchrony strength is summarized over the quantification window in Fig. 4d, exhibiting a monotonic relationship with LED intensity. Intriguingly, while we observe a bidirectional trend of the effect of the LED activation of L6CT neurons on the ongoing synchrony of the VPm population that parallels the trend of the VPm firing rate presented in Fig. 2, the changes are small and nonsignificant (Figure S6). We explore the magnitudes of the changes in L6CT population synchrony strength in subsequent analysis.

## Causal manipulation of L6CT population synchrony results in bidirectional modulation of ongoing firing rates in the VPm

To determine the contribution of L6CT population synchrony to the bidirectional modulation of ongoing VPm activity, in a subset of our recordings (Fig. 5a), we designed and presented LED inputs that varied the synchrony of the L6CT population while maintaining a fixed overall firing rate.

Figure 5b shows the spiking of two simultaneously recorded L6CT neurons in response to two continuous LED inputs: (1) the previously used ramp-and-hold, referred to as the 0 Hz LED input, or (2) a 250 ms ramp followed by uniformly distributed white noise modulation of the LED intensity with a maximum frequency content of 500 Hz, thus referred to as the 500 Hz LED input. For this example, frozen realizations of white noise were presented, such that the 500 Hz LED inputs are the same across trials. The 0 and 500 Hz LED inputs have

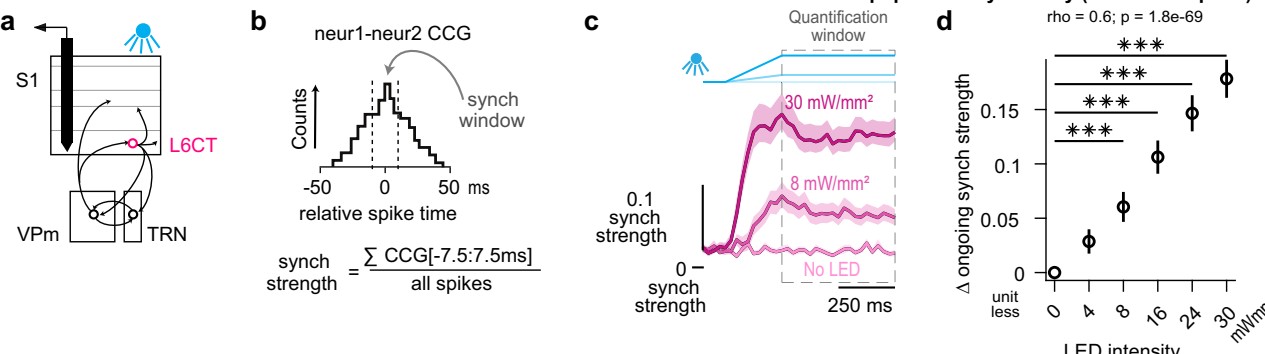

**Fig. 4 | Optogenetic inputs of increasing LED intensity monotonically enhance the synchrony of the L6CT population. a** Further assessment of the direct effect of the LED on L6CT neuron activity was measured from the same in vivo, silicon probe recordings of S1 neuronal spiking used in Fig. 1. **b** We assessed the L6CT population synchrony by measuring the strength which is defined for pairs of simultaneously recorded neurons and only includes spikes for a given L6CT neuron that occur within 7.5 ms of another L6CT neuron, normalized by the overall firing rates of the two neurons (see Methods). **c** Grand PSTHs of the L6CT population synchrony strength to 0, 8, and 30 mW/mm² LED inputs (mean +/− sem across

neuron pairs). The population's synchrony strength is also enhanced in response to increasing LED intensity. **d** The mean +/− sem change in synchrony strength of the L6CT population contributing to the PSTHs in *c* is monotonically enhanced by increasing LED intensities ($n = 117$ neuron pairs; 4 mW/mm²: $p = 0.62$; 8 mW/mm²: $p = 4.6e−5$; 16 mW/mm²: $p = 4.3e−18$; 24 mW/mm²: $p = 1.0e−19$; 30 mW/mm: $p = 4.1e−20$; $p$-value indicates pairwise comparisons to no LED condition using the two-sided Wilcoxon signed rank test with a Bonferroni correction; *** indicates $p < 2.0e−4$; rho & $p$ value: Spearman's rank correlation).

equivalent mean LED intensities and therefore similarly activate L6CT neurons, on average, across trials. However, in response to the 500 Hz LED input, the spike timing across L6CT neurons is often near coincident due to high-frequency events in the LED input. As a result, the rate of synchronous spikes between the pair of neurons is higher compared to that of the 0 Hz LED input.

To further probe the effects of L6CT population synchrony, we presented additional LED inputs consisting of lowpass filtered, non-frozen realizations of white noise with 10 and 100 Hz cutoff frequencies, referred to as the 10 and 100 Hz LED inputs, respectively; all four LED inputs have an equal mean of 16 mW/mm². Across the L6CT population, we find that the firing rate is not significantly changed by the 10, 100, and 500 Hz LED inputs compared to the 0 Hz LED input (Fig. 5c, d). However, there is a corresponding, significant increase in the synchrony strength for the noisy LED inputs as compared to the 0 Hz LED input (Fig. 5e, f). Notably, these synchrony strengths are within the range observed from the ramped LED steps in Fig. 4d.

Having now decoupled the L6CT population synchrony from the firing rate, we then measured the effect of the increase in L6CT population synchrony on VPm activity. In response to the 0 Hz LED input with a mean intensity of 16 mW/mm², the ongoing firing rate in the VPm population is suppressed relative to baseline, although this change is not statistically significant, as is also the case in Fig. 2f. However, even though the overall firing rate of the L6CT neuron population is the same, increased L6CT synchrony enhances the ongoing firing rate in the VPm population above baseline (Fig. 5g, h).

Taken together, the synchrony of the L6CT population plays an important role in contributing to the net modulatory effect of L6CT neuron activation on VPm ongoing activity (Figure S7). These shifts in L6CT synchrony can arise from overall changes in L6CT neurons' firing rates as well as from changes in the temporal precision of equivalent numbers of L6CT neurons' spikes.

**Modeling the role of excitatory and inhibitory conductances in the L6CT synchrony-dependent bidirectional modulation of VPm ongoing activity**

To understand the synaptic conductances underlying the L6CT synchrony-dependent bidirectional modulation of VPm, we constructed a simple circuit model that illustrates the key interactions between L6CT, VPm, and TRN neurons. In contrast to a full network population model matching the numbers of neurons and connections,

we followed the spirit of previous reduced complexity models developed for the thalamocortical circuit[50,51]. The circuit consists of two primary nodes capturing the average VPm and TRN population dynamics, each modeled with linear integrate-and-fire (LIF) dynamics. Each node receives common Poisson spike train inputs from 35 L6CT neurons with independently controlled firing rates and synchrony. Each node also receives independent, noisy, excitatory inputs to generate spontaneous spiking. Finally, the TRN node provides inhibitory input to the VPm node (Fig. 6a; see Methods).

Figure 6b shows single trial realizations of the model, where we assess the conductances and membrane potential in the VPm as we increase the synchrony of the L6CT spike trains while maintaining a constant firing rate. Here, the excitatory conductance arises from L6CT spiking and the inhibitory conductance arises from TRN spiking.

VPm spiking activity replicates our experimental results: a suppression of spontaneous spiking at low synchrony and an enhancement of spiking at high synchrony. Quantifying the average change in VPm firing rate across trials, we observe an L6CT synchrony-dependent bidirectional modulation of VPm activity across a continuum of L6CT synchrony strengths (Fig. 6c). Unsurprisingly, the model produces much less variability across trials than the experimental data. Notably, these synchrony strengths are within the range observed in Fig. 4d and reiterate that even seemingly small changes in measures of synchrony can have notable effects on the firing activity of downstream neurons[35].

To further understand the evolution of the membrane potential under varying levels of L6CT synchrony, we examined the underlying model conductances, an aspect that is hidden in our extracellular recordings. We found that under the low synchrony condition, the magnitude of the inhibitory conductance dominates over that of the excitatory conductance for the entire trial duration. Individual L6CT spikes cannot sufficiently activate the excitatory conductance to ultimately depolarize the membrane potential for spiking. Moreover, not only do individual L6CT spikes sufficiently activate the inhibitory conductance, but the inhibitory conductance never decays back to baseline due to the frequent occurrence of an L6CT spike. As a result, the VPm membrane potential is hyperpolarized and spontaneous spiking is suppressed (Fig. 6d). In contrast, in the high synchrony condition, the synchronous L6CT spikes result in an excitatory conductance that is sufficient to enable summation to spike before the onset of the disynaptic, inhibitory conductance.

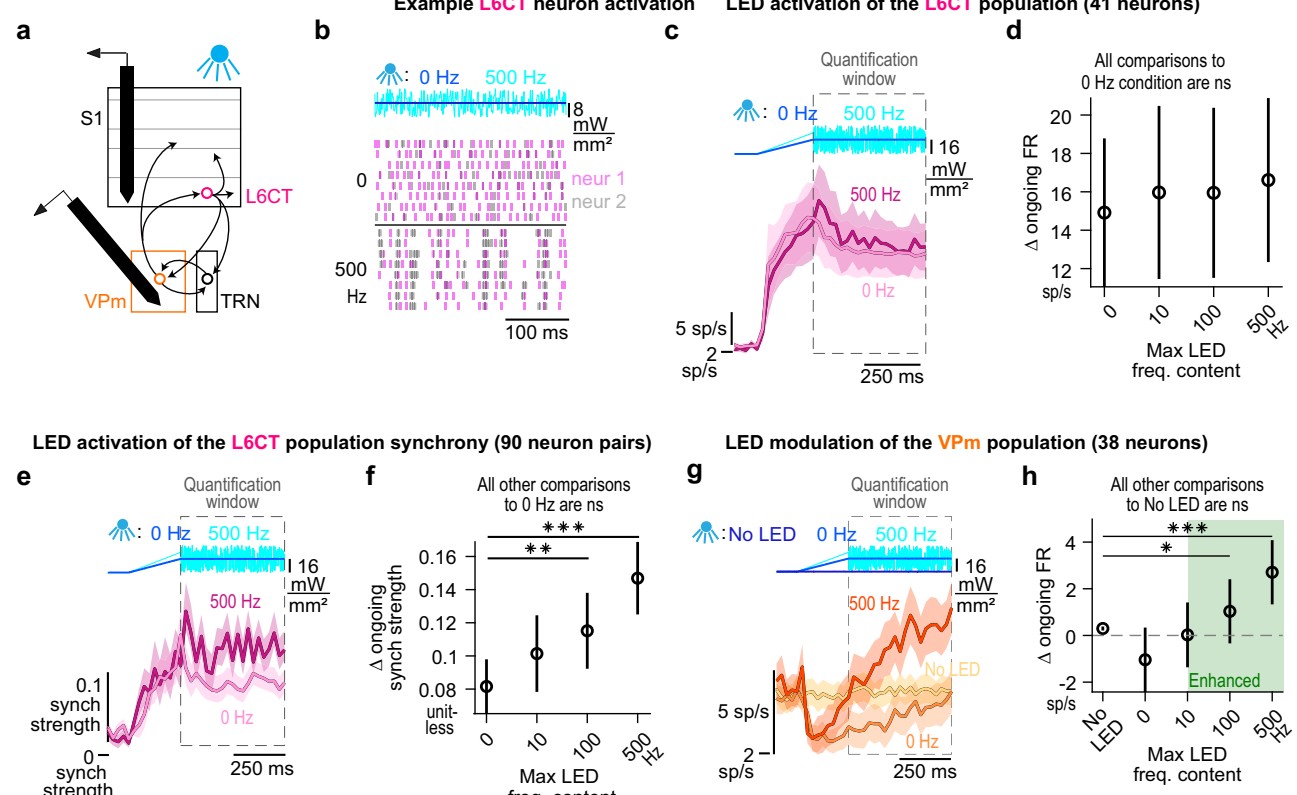

**Fig. 5 | Causal manipulation of L6CT population synchrony results in bidirectional modulation of ongoing firing rates in the VPm. a** The effects of additional LED inputs are measured from in vivo, silicon probe recordings of neuronal spiking, either in S1 or the VPm of awake, NTSR1-Cre mice. **b** Raster plots of two simultaneously recorded L6CT neurons activated by two frozen LED inputs of equal mean but varying maximum frequency content: 0 or 500 Hz. **c** Grand PSTHs of the L6CT population to the 0 and 500 Hz LED inputs (mean +/− sem across neurons). Note that the noise across trials for this analysis is not frozen. **d** The mean +/− sem change in ongoing firing rate of the L6CT population contributing to the PSTHs in (**c**) is constant in response to increasing frequency content of the LED input ($n = 41$ neurons; 10 Hz: $p = 0.12$; 100 Hz: $p = 0.12$; 500 Hz: $p = 0.12$; $p$ value indicates pairwise comparison to 0 Hz condition using the two-sided Wilcoxon signed rank test with a Bonferroni correction). **e** Grand PSTHs of the L6CT population synchrony to

the 0 and 500 Hz LED inputs (mean +/− sem across neuron pairs). **f** The mean +/− sem change in synchrony strength of the L6CT population contributing to the PSTHs in **e** is enhanced by increasing frequency content of the LED epoch ($n = 90$ neuron pairs; 10 Hz: $p = 5.1e−2$; 100 Hz: $p = 4.7e−4$; 500 Hz: $p = 3.8e−9$; $p$ value indicates pairwise comparison to the 0Hz condition using the two-sided Wilcoxon signed rank test with a Bonferroni correction; ** indicates $p < 3.33e−3$; *** indicates $p < 3.33e−4$. **g** Grand PSTHs of the VPm population to the 0 and 500 Hz LED input activation of L6CT neurons (mean +/− sem across neurons). **h** The mean +/− sem change in ongoing firing rate of the VPm population contributing to the PSTHs in (**g**) is enhanced in a graded manner by the LED frequency content ($n = 38$ neurons; 0 Hz: $p = 0.85$; 10 Hz: $p = 0.9$; 500 Hz: $p = 3.5e−5$; p-value indicates pairwise comparison to No LED condition using the two-sided Wilcoxon signed rank test with a Bonferroni correction; * indicates $p < 1.25e−2$; *** indicates $p < 2.5e−4$).

Several well-documented factors undoubtedly play a role in our experimental findings such as the short-term dynamics of the involved synapses[5] as well as the fact that L6CT inputs result in modulatory NMDA plateau potentials in the distal dendrites of FO thalamic neurons[52]. We propose that the interplay of both the relative strength and timing of the excitatory and inhibitory conductances, as observed in our model, is a relatively simple mechanism that plays an additional key role in our experimental findings. In this scenario, the synchrony of the L6CT neuron population strongly influences the net modulatory effect of L6CT neuron input on its VPm neuron target, thus serving as an important factor in L6CT function.

### Optogenetic activation of L6CT neurons enhances VPm sensory responses

Having established that different levels of activation of L6CT neurons cause bidirectional changes in the ongoing firing rate in VPm neurons, we next sought to understand the effect of L6CT neuron activation on sensory responses evoked in the VPm. We recorded the spiking response of populations of individual VPm neurons to concurrent single whisker deflection and optogenetic L6CT input of increasing LED intensity (Fig. 7a).

Figure 7b, c show the response of an example VPm neuron to a sawtooth whisker stimulus occurring 750 ms after the onset of the LED activation of L6CT neurons. As previously noted, the effect of the optogenetic activation of L6CT neurons on ongoing activity in VPm neurons varies with LED intensity, with suppression at 8 mW/mm² and enhancement at 30 mW/mm² (Fig. 7b, left panel). Taking the direction of change in ongoing firing rate as a cue for the general excitability of VPm neurons, a simple prediction would be that the sensory response in VPm neurons would be amplified when VPm background activity is increased and attenuated when VPm background activity is suppressed. However, as qualitatively highlighted by the sensory response of the example VPm neuron in Fig. 7b (right panel) and the VPm population in Fig. 7c, despite the bidirectional modulation of the ongoing activity, the sensory response is not bidirectionally modulated by the LED inputs.

We accounted for both the pre- and post-sensory stimulus modulatory effects of L6CT activation and quantified the sensory response by measuring the trial-averaged stimulus-evoked rate subtracted from the pre-stimulus rate, in equivalent 30 ms duration windows. Across the entire VPm population, increasing optogenetic activation of L6CT neurons tends to enhance the magnitude of the sensory response

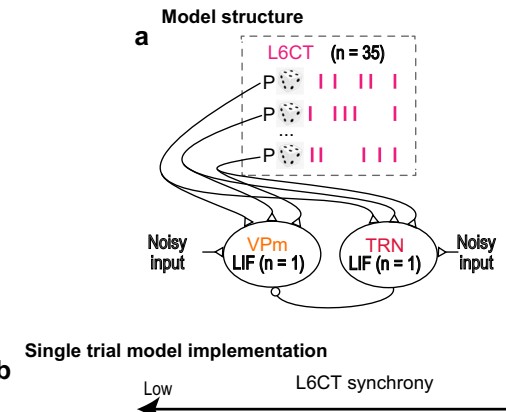

**Model structure**

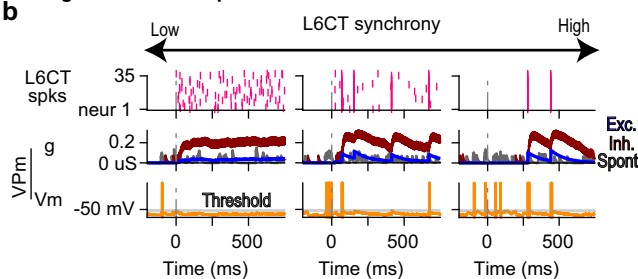

**Single trial model implementation**

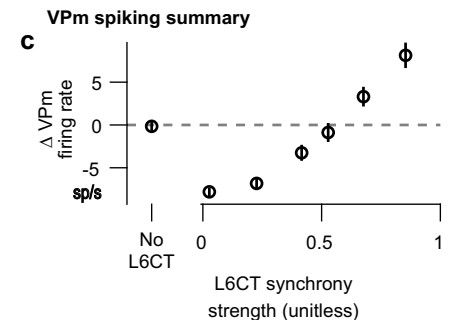

**VPm spiking summary**

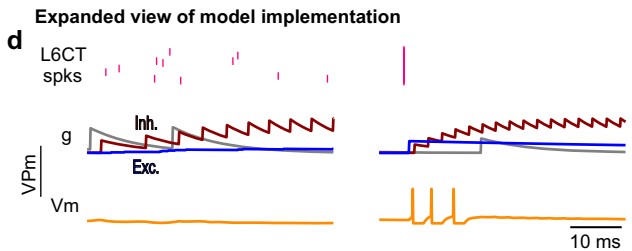

**Expanded view of model implementation**

**Fig. 6 | Modeling the role of excitatory and inhibitory conductances in the L6CT synchrony-dependent bidirectional modulation of VPm ongoing activity. a** The model structure consists of representative VPm and TRN neurons modeled as leaky, integrate and fire (LIF) neurons, each integrating common L6CT spike train inputs with controlled synchrony and firing rate. The VPm neuron integrated inhibitory input from the spiking of the TRN neuron. Spontaneous activity for each LIF neuron was driven by noisy, excitatory inputs. **b** Single trial model simulations run by holding the L6CT neurons' firing rates fixed while varying the synchrony of the L6CT inputs result in a shift from suppression to enhancement of VPm spiking in a graded manner with increasing synchrony. Note that the spontaneous conductance has been scaled 5x for better visualization. **c** The mean (+/− sem across 50 trials) change in VPm firing rate with increasing L6CT population synchrony, but constant L6CT neurons' firing rate (2.5 sp/s) replicates the experimentally observed, L6CT synchrony-dependent, bidirectional modulation of VPm. **d** An expanded view of the single trial components and underlying conductances in VPm highlights the key factors underlying L6CT synchrony-dependent, bidirectional modulation of VPm: (1) the relative strength of the inhibitory conductance compared to the excitatory conductance, and (2) the relative timing of the monosynaptic excitatory and disynaptic inhibitory conductances.

(Fig. 7d, e). These effects are measured in trials without whisker movement, as presented in Fig. 7. The effect in trials with whisker movement is associated with a reduction of the overall magnitude of the VPm sensory response as well as a reduction in the measured effect of L6CT activation on the VPm sensory response (Figure S8). We also observe a corresponding increase in the sensory response synchrony strength of the VPm population response across all LED intensities (Figure S9).

Interestingly, the VPm population sensory response is not further affected when we present the sensory stimulus during the 500 Hz noise LED input, as compared to the 0 Hz LED step of equal mean. This suggests that a selective increase in the synchrony of the L6CT inputs does not further impact the sensory responses (Figure S10), in contrast to what is observed for ongoing VPm activity (Fig. 5).

To assess the effect of L6CT modulation on the transfer of sensory inputs through VPm in a slice preparation, Crandall et al. co-activated L6CT neurons and medial lemniscus (ML) axons. They found a clear relationship between the effect of L6CT activation on the response of VPm neurons to ML stimulation and the biphasic modulatory effect on spontaneous VPm spiking[5]. From such a study, it is reasonable to expect that VPm sensory responses would be suppressed in conditions where VPm ongoing activity is suppressed, at 4 and 8 mW/mm². Yet, we rather observe that neurons with increasingly suppressed ongoing activity tend towards a more enhanced sensory response compared to the no LED condition (Figure S11). While one might expect an increase in T-type calcium channel-mediated burst spikes in conditions where the VPm ongoing activity is suppressed[15,33], we do not observe any significant changes in stimulus-evoked VPm bursting upon L6CT activation (Figure S12). Thus, the observed bidirectional changes in VPm ongoing activity do not trivially predict the VPm sensory response magnitudes, as sensory responses are consistently amplified by L6CT activation.

## Optogenetic activation of L6CT neurons dynamically modulates activity in S1

Given the effect of optogenetic activation on ongoing and stimulus-evoked activity in VPm neurons, it is important to determine how the cortico-thalamo-cortical modulatory effects coupled with modulatory effects directly from L6CT intracortical projections to ultimately impact S1. We recorded the spiking response of populations of individual S1 neurons across all cortical layers while presenting the same optogenetic and tactile inputs presented in Fig. 7 (Fig. 8a). All neurons were identified as either putative excitatory regular-spiking (RS) or inhibitory fast-spiking (FS) neurons based on the trough-to-peak time of their spike waveform widths (Fig. 8b; See Methods).

Figure 8c, d show an example S1 RS (non-L6CT) neuron's response to optogenetic activation of L6CT neurons at varying LED intensities with an embedded sensory stimulus. Pre-stimulus, the LED clearly suppressed the ongoing spiking activity of this neuron. Intriguingly, more of the spiking was suppressed at 8 mW/mm² than at 30 mW/mm². As shown in the right panels of Fig. 8c, d, the sensory response of this neuron was also modestly suppressed by optogenetic activation of L6CT neurons.

Both when looking at RS and FS sub-populations across all layers (Fig. 9a, b), as well as when parsing neuronal responses by layers (Figure S13 & S14), optogenetic activation of L6CT neurons leads to a suppression of ongoing activity in both S1 sub-populations, with FS neurons suppressed to a larger extent. This suppression likely arises in large part from L6CT direct intracortical projections to S1 inhibitory neurons, as previously characterized in the visual cortex[53].

While there is a clear suppression of both sub-populations at 8 mW/mm², there is no further suppression at higher LED intensities. Rather, in both sub-populations, there is a small reversal of the suppressive effects beyond 16 mW/mm², reminiscent of the observed changes in ongoing VPm activity that would then be propagated to S1 (Fig. 9c). To assess this

**Sensory response of an example VPm neuron**

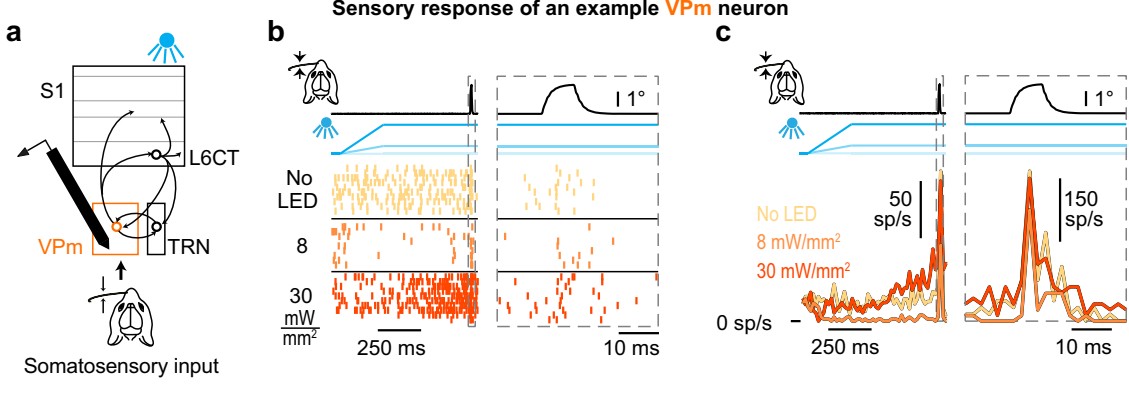

**Sensory response of the VPm population (65 neurons)**

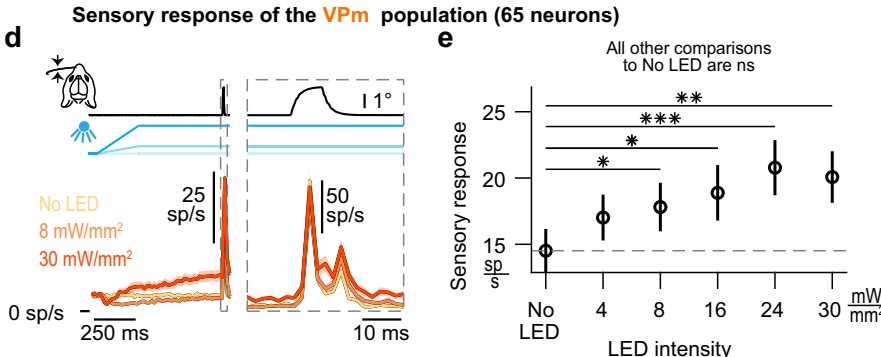

**Fig. 7 | Optogenetic activation of L6CT neurons enhances VPm sensory responses. a** The effect of L6CT neuronal activation on sensory responses in VPm is measured from recordings of neuronal spiking in the VPm while concurrently deflecting a single whisker for somatosensory input and increasing optogenetic activation of L6CT neurons in awake, NTSR1-Cre mice. **b** Raster plots of an example VPm neuron's sensory response with concurrent optogenetic activation of L6CT neurons at 0, 8, and 30 mW/mm². This neuron's ongoing spiking activity is bidirectionally modulated and its stimulus-evoked response to sawtooth, somatosensory input is enhanced by increasing optogenetic activation of L6CT neurons. The right panel is an expanded view of the same neuron's sensory response. **c** PSTHs of the same neuron's spiking in (**b**). Note the change in the vertical scale in the right and left views due to differing bin sizes. **d** Grand PSTHs of the VPm neuron population's sensory response with concurrent optogenetic activation of L6CT neurons at 0, 8, and 30 mW/mm² (mean +/− sem across neurons; n = 65 neurons). The population's ongoing activity is bidirectionally modulated, but the stimulus-evoked response is enhanced by increasing optogenetic activation of L6CT neurons. The right panel is an expanded view of the same population's sensory response. Note the change in the vertical scale in the right and left views. **e** Optogenetic activation of L6CT neurons increases the mean, baseline-subtracted, stimulus-evoked response of the VPm population contributing to the PSTHs in *d* (+/− sem; n = 65 neurons; 4 mW/mm²: $p = 8.0e{-}3$; 8 mW/mm²: $p = 9.7e{-}4$; 16 mW/mm²: $p = 3.7e{-}4$; 24 mW/mm²: $p = 1.3e{-}5$; 30 mW/mm²: $p = 9.5e{-}5$; $p$-value indicates pairwise comparison to No LED condition using the two-sided Wilcoxon signed rank test with a Bonferroni correction; * indicates $p < 1.0e{-}2$; ** indicates $p < 2.0e{-}3$; *** indicates $p < 2.0e{-}4$).

reversal, we made pairwise comparisons between the 8 mW/mm² condition and all other LED-on conditions. While the reversal is small in the RS sub-population, the change in ongoing firing rate in the FS population is significantly higher at 24 and 30 mW/mm² compared to the change at 8 mW/mm² (Fig. 9d). Such significance is also observed in FS populations in layers 4 and 5, such that the change in ongoing activity in response to the 30 mW/mm² condition is statistically unchanged from baseline compared to the clear suppression observed in response to the 4 and 8 mW/mm² LED inputs (Figure S13b). Despite the well-documented effects of behavioral state changes reflected in whisker movement on overall cortical spiking[54], whisker movement was associated with only a modest decrease in the measured effect of L6CT activation on ongoing RS activity and was not associated with any change in the effect of L6CT activation on ongoing FS activity (Figure S15).

We then characterized the effect of optogenetic activation of L6CT neurons on sensory responses of the S1 sub-populations. While there is an overall small decrease in the RS (non-L6CT) population sensory response between the no LED condition and LED-on conditions (less than 1 sp/s; Fig. 9e), the sensory response for most S1 RS (non-L6CT) neurons is statistically unchanged, such that all pairwise comparisons of the RS population's sensory response between no LED conditions to LED-on conditions of increasing intensities are nonsignificant (Fig. 9f). Whisker movement is associated with an overall decrease in RS sensory responses across all LED intensities. Further, pairwise comparisons of the RS population's sensory response between no LED and LED-on conditions are also nonsignificant in trials with whisker movement (Figure S16). When parsing RS sensory responses by layer, the only clear trend that emerges is in layer 6, where responses are suppressed (Figure S17b).

Interestingly, the sensory response of FS neurons at 4 mW/mm² is significantly higher than baseline conditions, whereas the response at 30 mW/mm² is non-significantly suppressed. Given the observed maximal suppression of ongoing FS activity at 8 mW/mm² as well as the apparent reversal in the effect of LED intensity on sensory response, we again compared the sensory response of all LED-on conditions to that at 8 mW/mm². Under this analysis, the sensory response at 30 mW/mm² is significantly lower than the response at 8 mW/mm² (Fig. 9f). Features of these findings are present when parsing the neuronal response by layer, with significance in layers 4 and 5 (Figure S18). Despite an overall decrease in sensory response across all LED intensities in trials with whisker movement, the trend of an enhanced response at 4 mW/mm² and suppressed response at higher intensities is also present but nonsignificant in trials with whisker movement (Figure S16).

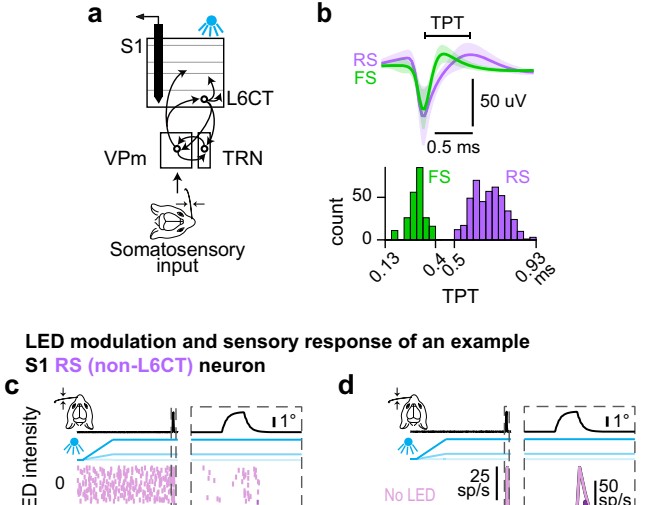

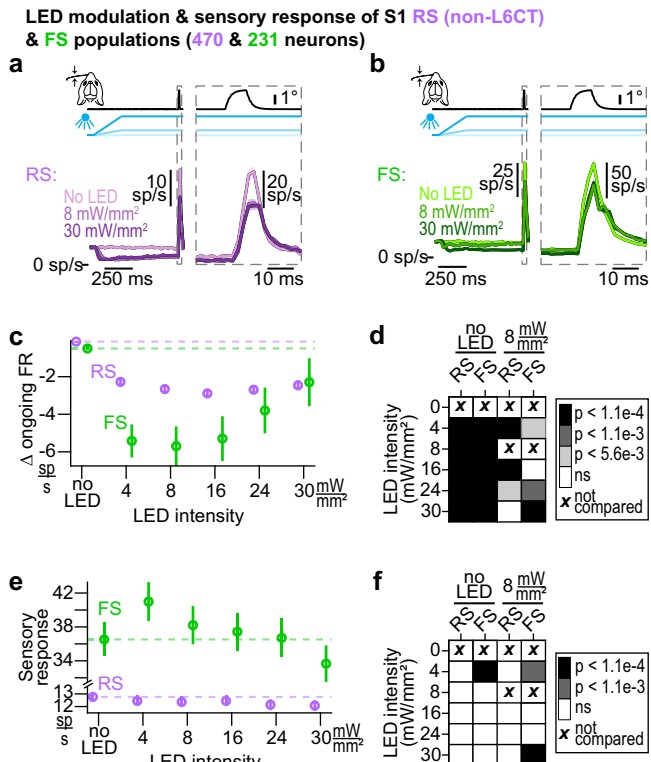

**Fig. 8 | The effect of optogenetic activation of L6CT neurons on S1 activity.**
**a** The effect of L6CT neuron activation on ongoing activity and sensory responses in S1 is measured from recordings of neuronal spiking in S1 while concurrently deflecting a single whisker for somatosensory input and increasing optogenetic activation of L6CT neurons in awake, NTSR1-Cre mice (*n* = 36 recordings in 11 mice). **b** Mean waveform +/− standard deviation across S1 neurons identified as either regular spiking (RS) or fast spiking (FS) based on the trough-to-peak time (TPT) of their spike waveform (RS > 0.5 ms and FS < 0.4 ms; see Methods). **c** Raster plots of an example RS (non-L6CT) neuron's sensory response with concurrent optogenetic activation of L6CT neurons at 0, 8, and 30 mW/mm². This neuron's ongoing and stimulus-evoked spiking is suppressed by concurrent optogenetic activation of L6CT neurons (left panel). The right view is an expansion of the same neuron's sensory response. **d**. PSTHs of the same neuron's spiking in *c*. Note the change in the vertical scale in the right and left views due to differing bin sizes.

The net observed effect of corticothalamic feedback on S1 is a sum of both L6CT-mediated changes through the cortico-thalamo-cortical route of modulation as well as through direct intracortical projections. This complexity underlies our results. While the overall effect of optogenetic activation of L6CT neurons on ongoing activity in both S1 subpopulations is suppressive, the effect on the sensory responses of the FS sub-population is dynamic, such that there is a sweet spot of enhancement. These dynamics would then contribute to the observed response in the RS population.

## Discussion

In this study, we conducted a functional dissection of the somato-sensory cortico-thalamo-cortical circuit by way of multi-region elec-trophysiology, strategic design of optogenetic manipulations, and sensory stimulation, as well as a simple circuit model. In doing so, we determined the direct effect of L6CT neurons on thalamocortical sensory processing across a range of activation patterns during wakefulness. We expanded on previous in-vivo findings that provide evidence for L6CT neurons as static modulators that either suppress or enhance FO thalamus[7–14,55] and establish L6CT feedback as a dynamic modulator that can both suppress and enhance the ongoing spiking of a VPm neuron. Further, we pinpoint that the bidirectional modulation of VPm is mediated both by differential sensitivity of L6CT and TRN neurons to increasing LED inputs as well as specific aspects of the activity patterns of the L6CT population: the firing rate and synchrony, properties that, generally, vary in the cortex across behaviors[54]. In addition, we provide evidence for the effect of this dynamic modulation on S1 neurons and find non-monotonic changes in S1 RS and FS

**Fig. 9 | Optogenetic activation of L6CT neurons dynamically modulates activity in S1. a** Grand PSTHs of the population of S1 RS (non-L6CT) neurons' sensory response (mean +/- sem across neurons from all layers). The right view is an expansion of the population sensory response; note the change in the vertical scale in the right and left views. **b** Same as (**a**), but for the FS population. **c** The mean +/− sem change in ongoing activity of the S1 RS (non-L6CT) and FS sub-populations contributing to the PSTHs in a (*n* = 470 RS neurons, 231 FS neurons). Both sub-populations are suppressed by all LED inputs, although to different extents.
**d** Confusion matrix indicating statistical significance in the change in ongoing firing rate observed in *c* when comparing to both no LED conditions as well as to 8 mW/mm². Color indicates *p* value using the two-sided Wilcoxon signed rank test with a Bonferroni correction (RS/FS population p-values when comparing to no LED condition: 4 mW/mm²: 9.8e−71/2.6e−18; 8 mW/mm²: 1.7e−85/1.2e−19; 16 mW/mm²: 5.3e−90/3.9e−17; 24 mW/mm²: 3.2e−87/1e−10; 30 mW/mm²: 4.8e−76/1.7e−5. RS/FS population *p* values when comparing to 8 mW/mm² condition: 4 mW/mm²: 1e−19/6e−4; 16 mW/mm²: 1.1e−8/8.9e−1; 24 mW/mm²: 2.2e−5/2.8e−4; 30 mW/mm²: 1.3e−1/5.5e−7.). Both sub-populations are suppressed across LED intensities compared to no LED levels. Also, the change at 8 mW/mm² is lower than at 4, 16, and 24 mW/mm² for the RS sub-population and 4, 24, and 30 mW/mm² for the FS sub-population. **e** Same as (**c**), but for sensory responses (*n* = 470 RS neurons, 231 FS neurons). **f** Same as (**d**), but for sensory responses (RS/FS population p-values when comparing to the no LED condition: 4 mW/mm²: 9.2e−2/2.8e−7; 8 mW/mm²: 5.6e−3/6.2e−3; 16 mW/mm²: 5.8e−3/3.3e−2; 24 mW/mm²: 1e−1/7.3e−1; 30 mW/mm²: 8.9e−3/1.9e−1. RS/FS population p-values when comparing to 8 mW/mm² condition: 4 mW/mm²: 3.9e−1/1.7e−4; 16 mW/mm²: 8.6e−1/4.6e−2; 24 mW/mm²: 1.4e−2/1.9e−2; 30 mW/mm²: 4.1e−1/1.3e−7.). While there is no significant change in sensory response in the RS sub-population, the FS sub-population sensory response is significantly enhanced at 4 mW/mm² compared to no LED conditions. Further, compared to the response at 8 mW/mm², the response at 4 mW/mm² is sig-nificantly enhanced and the response at 30 mW/mm² is significantly suppressed.

ongoing activity as well as FS stimulus-evoked activity. Collectively, L6CT neurons play a role in the continuous and context-dependent modulation of thalamocortical sensory signaling through motor cor-ollary signals, arousal, and other behavioral states[21,25–28], and we pro-vide comprehensive evidence for the effect of the L6CT feedback mechanism across such facets.

Much of what we know about this complex circuit has been learned from studies conducted in ex-vivo brain slice preparations

examining L6CT feedback at the cellular level[1–6,19,52]. The consensus from these studies is that synaptic input from L6CT projections serves to modulate the excitability of FO thalamic neurons. But conflicting results have pointed to both excitatory and suppressive effects on excitability. Importantly, in an ex-vivo slice preparation, Crandall et al. found that L6CT neurons can switch their influence on each VPm neuron from excitatory to suppressive depending on the frequency of stimulation[5]. The general nature of those results is supported here, but the slice experiments pointed to modulation that is mediated by short-term synaptic plasticity. The L6CT synchrony effects demonstrated here likely work synergistically with the synaptic plasticity mechanisms at the thalamo-reticular targets to provide a wide and continuous range of activity-dependent, top-down modulation of VPm.

The strong activation of TRN at low intensities likely also plays a role in the bidirectional modulation of VPm. This couples with the rate of increased L6CT spiking compared to the rate of increased TRN spiking with increasing LED intensity. More specifically, the strong activation of TRN at even relatively low levels of optogenetic activation of L6CT (4 mW/mm$^2$) could explain an overall suppression of VPm spiking. But the effect of increasing LED intensity on L6CT versus TRN spiking suggests the excitatory conductance to VPm is likely increasing at a relatively faster rate than the inhibitory conductance with increased LED intensity, the net sum of which would contribute to the observed bidirectional modulation of VPm.

As suggested by the circuit model in this study, the L6CT synchrony-dependent, bidirectional modulation of VPm emerges, first, because of the strong inhibitory conductance from TRN inputs relative to the strength of the direct excitation from L6CT neurons. The second important feature is the circuit architecture that enables a monosynaptic excitatory conductance and an inherently delayed, disynaptic inhibitory conductance in the VPm. These elements are key to the window of opportunity classically described in the feedforward path for FO thalamic inputs onto layer 4 cortical RS neurons[44,45,50]. The premise of this mechanism is that while the strength of individual thalamocortical synaptic inputs is modest, numerous inputs converge on cortical synaptic targets, making the cortical neurons highly sensitive to the synchrony of the ascending inputs[44]. Indeed, L6CT inputs to VPm and TRN are numerous and highly convergent[1–3,18], contributing to a synchrony-dependent effect of this classically modulatory circuit. This would provide another example to support the possibility of a synchrony-driven window of opportunity as a canonical circuit computation in the brain.

It is, however, important to note that there are several fundamentally different biological considerations between L6CT and thalamocortical inputs that would impact the window of opportunity in the feedback versus feed-forward circuits. First, L6CT neurons have a wide range of axon conduction latencies relative to that of thalamocortical projections. This would impact the synchronization of the postsynaptic conductances towards summation in the sub-cortical targets. It is worth noting that latencies for corticothalamic signal propagation in the range of 1–7 ms have been measured amongst the full range of axon conduction latencies[56]. This would lead to a variability that is relatively small compared to the 15 ms (+/−7.5 ms) window of opportunity that we consider here. Nonetheless, the full impact of the heterogeneity of L6CT conduction velocities in our study is currently unknown and should be pursued in future studies. In addition, it is well known that thalamocortical inputs produce fast conductance changes on the proximal dendrites of their postsynaptic targets via ionotropic receptors. However, L6CT neurons activate both ionotropic and metabotropic receptors, the latter of which produces prolonged NMDA plateau potentials in the distal dendrites of FO thalamic neurons[57]. While the time constants of the L6CT synapses in our model were adjusted to reflect the longer timescale of NMDA plateau potentials (Table 1), the nature of our single-compartment model precludes capturing such biological complexity. It is nonetheless important to note that such factors undoubtedly contribute to our experimental results and would impact the window of opportunity. Also, both the L6CT synapses onto VPm and TRN as well as the TRN synapses onto VPm undergo short-term synaptic plasticity[5]. None of these factors were implemented in the circuit model in this study because our goal was to construct the simplest possible model to explain the observed phenomena. Undoubtedly though, the dynamics of these synapses are changing in the experimental conditions utilized here and likely complement the demonstrated synchrony-dependent effects. Finally, VPm and TRN neurons have connectivity between each other, in addition to recurrent connectivity within TRN. While the initial transient response to the optogenetic activation engages the full range of connectivity in this circuit, the later steady-state response appears to be dominated by L6CT-VPm and L6CT-TRN-VPm projections. In the model, the TRN-VPm influence likely reflects the combined effects of the omitted circuit elements that together set the overall inhibitory tone and dictate the window of opportunity.

The very nature of the reciprocal connectivity of the thalamus and cortex makes assigning causal influences challenging. While optogenetic interventional approaches in the primary visual cortex have shown that the suppression of the ongoing activity of both the RS and FS S1 subpopulations arises predominantly from direct intracortical L6CT projections to inhibitory translaminar cells[53], the cortical sensory response reflects the net culmination of two sub-circuits: the intracortical circuit and the cortico-thalamo-cortical circuit. The net effect of the L6CT activation on ongoing, baseline activity of both RS and FS neurons across cortical S1 is suppressive. The larger suppression of the cortical FS neurons relative to RS points to a change in E/I balance and a corresponding increase in cortical excitability with L6CT activation. Although the baseline-subtracted sensory-evoked response of the cortical RS neurons showed no apparent change with L6CT activation, the suppression of ongoing activity could result in a more potent effect of this population activity on downstream targets through a decrease in levels of synaptic depression and/or changes in the signal-to-noise ratio of the sensory stimulus.

Of the most dramatic effects shown here is the effect of the L6CT activation on S1 FS neurons, which exhibits a sweet spot of an enhanced cortical sensory-evoked response, which would then additionally inhibit cortical RS neurons. The result of all these sub-circuits being modulated by L6CT neurons is an RS sensory response magnitude that is surprisingly invariant to the manipulation. However, the effects of the differential modulation of the RS and FS subpopulations could have a profound impact on spatiotemporal sensory representations. Additional experimental investigation would be required to fully disentangle the contributions of each cortico-thalamo-cortical and intracortical sub-circuits, such that contributions from each individual component could be selectively and reversibly toggled. Moreover, while overall firing rates in RS neurons were unchanged, it is possible that L6CT activation could dynamically modulate other, more context-dependent aspects of RS neurons, such as their encoding of deviant[55] or adapting[10] stimuli, as has been explored in recent studies.

The experimental design likely has some effects on the observations made in this study. First, the sensory stimulus consisted of a brief, punctate, whisker deflection. The impulse-like nature of the stimulus is advantageous for establishing the input-output properties within a circuit. Moreover, the punctate stimulus is akin to stick-slip events that occur when a mouse palpates a surface, and thus is a relevant stimulus for probing sensory processing in the whisker pathway[58,59]. However, it is entirely possible that the effect of L6CT neuron activation on a more sustained and persistent sensory stimulus would be quite different. In contrast to the punctate sensory inputs utilized here that drive synchronous VPm spiking which would then strongly drive S1, sequences of whisker deflections such as those occurring during texture or curvature discrimination[58,60] or continuous noisy whisker inputs could

result in more measurable modulation of the sensory response of cortical RS neurons than observed here.

Second, the optogenetic activation of the L6CT neurons was achieved through LED illumination of the cortical surface in a broad and diffuse manner. While L6CT neuron projections are topographically organized, TRN projections are not, which results in modulatory effects in the FO thalamus that depend on topographic alignment[7,14]. However, the spiking activity of L6CT neurons has been shown to be modulated by extra-sensory features[21,25,26]. In those cases, the activation of L6CT neurons would likely be global, or at least extend to multiple barrel columns within S1, which is the type of conditions that the optogenetic manipulation here would represent. Moreover, the increase in L6CT neuron spiking would likely be long-lasting, more akin to sustained, long-duration manipulation of activation here, as opposed to optogenetic pulsing or analysis of epochs after the optogenetic activation of L6CT neurons utilized in other studies[9,15]. Such experiment designs likely modulate the circuit in an entirely different manner and explain differences in our study and others, such as a lack of changes in FO thalamic bursting observed here.

Finally, our experiments were conducted in head-fixed, stationary mice, such that the natural activity range of L6CT neurons would preclude that arising from many extra-sensory behavioral states. Our study lays the foundation of the direct effect of L6CT feedback on thalamocortical activity across a range of L6CT activity. Future studies must continue to characterize L6CT neurons (both their firing rates and synchrony) across a variety of behaviors and then manipulate L6CT neurons to causally assess the contribution of L6CT neuron spiking to these behaviors. Although architectures involving loops provide challenges to assigning causality, parsing observations based on behavioral/brain states and/or causal manipulation through interactive, closed-loop adaptive experimental strategies[61,62] may provide a path to disentangle complex circuits such as these involved in sensorimotor behaviors.

In our world which is full of dynamically changing sensory inputs, it is crucial for survival to be able to flexibly adjust our responsiveness to these inputs. Under contexts different from that studied here, Guo et al. found that the dynamic regulation facilitated by L6CT neurons in the FO thalamus mediates a trade-off between the detectability and discriminability of auditory stimuli[9]. Indeed, the dynamic effect of L6CT neurons and their location relatively early in the sensory processing hierarchy positions L6CT neurons to play critical roles in sensory processing, bidirectionally modulating activity in a way that would serve a variety of behavioral needs. Further, in addition to sensory circuits, the L6CT neuron circuit motif is also present in motor systems[63], potentially pointing to a canonical circuit motif that subserves dynamic neural signaling in complex environments.

## Methods

### Experimental model and subject details
All experiments were conducted in 11 male and 7 female NTSR1-cre (B6.FVB(Cg)-Tg(Ntsr1-cre)GN220Gsat/Mmcd, MMRRC) adult mice as well as 1 female wildtype, C57BL/6J mouse, all aged between 6 weeks and 6 months at the start of experimentation. The mice were housed under a reversed light-dark cycle and temperatures were set to 65-75 °F with 40-60% humidity. All procedures were approved by the Institutional Animal Care and Use Committee at the Georgia Institute of Technology and were in agreement with guidelines established by the National Institutes of Health.

### Headpost implantation
The mice were kept under isoflurane anesthesia (5% vaporized in O2 for induction, 1–1.5% for maintenance), their temperature maintained at ~37 C, and ophthalmic ointment (Puralube, Fisher Scientific) was applied to the eyes to prevent drying. A stereotaxic frame maintained the animal's head level in all three planes while a custom stainless steel headpost and recording chamber was affixed to the exposed skull with dental cement (C & B Metabond Parkell, Inc). The mice were then provided at least three days to recover[33,35,62,64,65].

### rAAV injection
An image of superficial vasculature was overlaid with a map of the whisker columns of the primary somatosensory cortex (S1), generated functionally via intrinsic signal optical imaging performed through a thinned skull under 1-1.5% isoflurane anesthesia[34,65,66]. On a subsequent day, also under 1–1.5% isoflurane anesthesia, 800 nL of the cre-dependent AAV5-EF1a-DIO-hChR2(H134R)-eYFP.WPRE.hGH[12,15] (UPenn Vector Core) was injected at a depth of 800 um in the C1 whisker column. The exposed skull was covered with a silicone elastomer (Kwik-Cast, WPI, Sarasota, FL) and the mice were returned to their home cage and provided at least three days to recover. Electrophysiological recordings were conducted no sooner than 3 weeks after the injection to allow for stable opsin expression.

### Habituation to head-fixation
Mice were habituated to head-fixation for a minimum of 5 days. Head-fixation sessions started with a duration of 15 min and went up to 90 min. Sessions gradually increased in duration from day-to-day as did the exposure to experimental conditions (e.g., whisker stimulation, exposure to optogenetic stimuli, and paw restraint). Throughout each session, mice were periodically rewarded with sweetened condensed milk[33,35,62,64,65].

### Electrophysiological recordings
All electrophysiological recording sessions were conducted in darkness using silicon probes in awake, head-fixed mice periodically rewarded with sweetened condensed milk[65]. Often, sessions consisted of simultaneous recording from two brain regions. Additionally, sessions were conducted no sooner than two hours after craniotomy procedures under 1-1.5% isoflurane anesthesia where the dura was left intact. No more than four craniotomies were made per animal.

The location of craniotomies to target S1 was based on the map generated via intrinsic signal optical imaging (see above). Recordings of the activity of S1 neurons were conducted using 32 or 64 channel silicon probes (A1x32-5 mm-25-177-A32, A1x32-Poly3-5mm-25s-177-A32, A1x64-Poly2-6mm-23s-160, or A1x64-Edge-6mm-20-177-A64, Neuronexus, Ann Arbor, MI) inserted to depths between 1250 and 1400 um at an angle of 35 degrees from vertical[65].

Craniotomies to target neurons in the ventral posteromedial (VPm) nucleus of the thalamus were made 2 mm posterior and 2 mm lateral from bregma. Often, a brief electrophysiological recording was first conducted under isoflurane anesthesia to verify the craniotomy location, or to extend the craniotomy as necessary. Recordings of VPm neurons were conducted using 32-channel silicon probes (A1x32-Poly3-5mm-25s-177-A32, Neuronexus, Ann Arbor, MI) inserted in the vertical plane to depths between 3200 and 3600 um[33–35,62].

Craniotomies to target neurons of the thalamic reticular nucleus (TRN) were made 2.4 mm posterior, with a medial edge between 1.5 and 2.5 mm from bregma[42]. Often, a brief electrophysiological recording was first conducted under isoflurane anesthesia to locate TRN neurons. Recordings of TRN neurons were conducted using 32-channel silicon probes (Buzsaki32 or A1x32-Poly3-5mm-25s-177-A32, Neuronexus, Ann Arbor, MI) inserted in the vertical plane to depths between 2900 and 3200 um[42,67].

Continuous signals were acquired using either the Cerebus acquisition system (Blackrock Neurotech, Salt Lake City, Utah) or TDT RZ2 Bioamp Processor (Tucker Davis Technologies, Alachua, FL). Signals were amplified, filtered between 0.3 Hz and 7.5 kHz, and digitized

at either 30,000 Hz or 24,414.0625 Hz. For each animal, we conducted one recording session per day and up to three recording sessions per craniotomy.

## Optogenetic stimulation

Optogenetic inputs were delivered using a 470 nm light emitting diode (LED, ThorLabs M470F3) coupled to a 400 um diameter optical fiber (ThorLabs M28L01) manually positioned on the skull surface near the recording probe. Command voltages for optogenetic inputs were controlled by custom scripts in MATLAB and Simulink Real-Time (MathWorks, Natick, MA), sampled at 1 kHz.

All optogenetic input began with a ramp of the LED intensity that was 250 ms in duration, serving to minimize transient neuronal effects from rapid increases in LED intensity as well as to minimize animal whisking[36]. In ramp-and-hold LED trials, the LED intensity was then held fixed at values ranging from 4 to 30 mW/mm² for 500–650 ms. Manipulations to vary L6CT synchrony included LED inputs that were 500–650 ms in duration labelled as either 500 Hz, 100 Hz, or 10 Hz trials. The input for 500 Hz trials consisted of uniformly distributed white noise. Inputs for the 100 Hz and 10 Hz trials consisted of uniformly distributed white noise that was low-pass filtered (3rd order Butterworth filter) at their respective frequencies. For all trials, the inter-trial interval was a minimum of 3.5 times the duration of the LED input with added uniformly distributed random jitter per-trial ranging from 0 - 0.1/0.5 ms.

## Whisker stimulation

For sensory stimulation, an individual whisker contralateral to the electrophysiological recording site was threaded into a narrow, 1.5 cm long tube connected to a precise, computer-controlled galvanometer[33–35,65] (Cambridge Technologies, Worthington, MN; sampled at 1 kHz). The whiskers were deflected at a velocity of 300 deg/s in the caudo-rostral direction with a sawtooth waveform consisting of rise and fall times of 8 ms, where velocity is defined by the average velocity over the rising phase. In trials with concurrent LED input and sensory stimulation, the sensory stimulus was presented 750 ms after the onset of the LED input.

## Videography and Whisker motion analysis

The movement of the whiskers ipsilateral to the electrophysiological recording site was captured under infrared illumination using CCD cameras (DMK 21BU04.H USB 2.0 monochrome industrial camera, The Imaging Source, LLC, Charlotte, NC; or EoSens CL MC1362, Mikrotron, Germany). Frames were individually triggered at 30 Hz or 200 Hz. Using FaceMap[68], whisking energy was extracted from manually defined regions of interest (ROIs) around the face. Using custom scripts, we computed the across-frame variation of the sum squared motion energy across ROIs. This time series was then smoothed and time points above a fixed threshold were classified as whisker motion epochs[35,65]. To measure the specific effects of our manipulations unconfounded by effects associated with whisking, all trials included for analysis of ongoing activity had no whisker motion occurring during the trial. For the analysis of sensory responses, only trials in which there was no whisker motion for 75 ms before and after the stimulus were included.

## Histology

For histological validation of the recording location, during one of the electrophysiological recording sessions per craniotomy, the probe was coated with DiI (0.2 mg/mL in ethanol, Invitrogen, Waltham, MA) to enable fluorescence imaging of the probe track[34,65,69]. In brief, after the final recording, mice were transcardially perfused with 1x PBS (137 mM NaCl, 2.7 mM KCl, and 10 mM PB, VWR) and then 4% PFA. The brains were extracted, post-fixed overnight in the PFA solution, and then sliced to 100-um thick sections on a vibratome (VT1000S, Leica

Biosystems Deer Park, IL). The slices were then incubated with DAPI (2 mM in PBS, AppliChem, Council Bluffs, Iowa) for 15 min, mounted on slides with a DABCO solution (1,4-Diazabicyclo[2.2.2]octane, Sigma), and imaged using a confocal microscope (Laser Scanning Microscope 900, Zeiss, Germany). Corresponding brain atlas sections were chosen manually (Franklin & Paxinos, 3rd edition).

## Spike sorting and identification of single neuron clusters

Individual recording sweeps were high-pass filtered (3rd order Butterworth filter; 500 Hz cut-off frequency), median-filtered across channels, and concatenated before being passed through Kilosort2 for automated spike sorting based on template-matching[70]. The data were then manually curated in Phy2[71]. With the exception of the control experiments conducted in a wildtype mouse, all neuron clusters considered for analysis passed two criteria: a signal-to-noise ratio of the mean spike waveform greater than three and less than 2% of all spikes violating a 2 ms imposed refractory period. Note that the use of more strict inclusion criteria does not change the nature of any claims presented here.

## Identification and classification of S1 neurons

All S1 neurons retained for analysis are required to be sensory stimulus-responsive, defined by passing a minimum of two out of three tests assessing if the stimulus caused a significant change in the spiking activity. The tests were conducted on spikes occurring in 50 ms a pre- and post-stimulus window. The first test was the two-sided Wilcoxon signed rank test on the pre- versus post-stimulus spike counts across trials. The second test determined whether zero overlapped with the 95% bootstrap confidence interval from the stimulus-evoked spike counts subtracted by the pre-stimulus spike counts. The final test used a post-stimulus peri-stimulus time histogram (PSTH) with 10 ms bins to determine whether a minimum of three bins were above or two bins were below the 95% bootstrap confidence interval of pre-stimulus activity[65].

S1 neurons were then classified as regular-spiking (RS) or fast-spiking (FS) based on the width of the mean spike waveform, defined as the trough-to-peak time[39] (TPT). Neurons with a TPT greater than 0.5 ms were classified as an RS neuron and neurons with a TPT less than 0.4 ms were classified as an FS neuron. Neurons with TPT widths between 0.4 ms and 0.5 ms were not analyzed[39,65,72].

Each neuron was then assigned to a layer relative to the center of layer 4, which was identified functionally from the sensory response on each channel of the recording probe. In brief, we extracted the trial-averaged sensory response for each channel across the following measures: threshold-crossings of the high-pass filtered recording trace, the local field potential (LFP), and the sources computed using current-source density analysis[72,73]. We extracted the channels with the earliest onset and largest response across these data types and the center of layer 4 was determined based on collective evidence. All channels, and therefore the neurons recorded on those channels, were then assigned to a layer based on their distance from the determined center of layer 4. Each layer was assigned the appropriate number of channels to cover the following thickness: 2/3: 300 um; 4: 200 um; 5: 300 um; 6: 300 um[65,72,74].

## Classification of L6CT neurons

To identify L6CT neurons, we used the stimulus-associated spike latency test (SALT) to statistically determine whether 5 ms, square LED pulses cause a significant increase in each neuron's spiking with low temporal jitter across trials[30]. In brief, SALT was used to compare the distribution of first spike latencies to the LED pulse compared to distributions created from spontaneous activity. Any deep, RS neuron with a p-value less than 0.01 was classified as a L6CT neuron. Note that L6CT neurons were not required to be sensory stimulus-responsive to be retained for analysis.

## Classification of VPm and TRN neurons

Neurons classified as VPm or TRN neurons are sensory stimulus-responsive neurons recorded at the relevant stereotaxic coordinate for each location. To determine whether a neuron is stimulus-responsive, we individually deflected multiple whiskers with a 10 Hz adapting train of sawtooth, punctate stimuli for one second. We extracted each neuron's response latency and magnitude to the first and last stimulus in the train in a 30 ms quantification window. VPm and TRN neurons were required to be significantly responsive to both the first and last stimulus in the train for any deflected whisker. Significance was defined using the two-sided Wilcoxon signed rank test to determine if the sensory stimulus caused a significant increase in the spike counts across trials compared to pre-stimulus (spontaneous) spike counts in an equivalent window. The neurons were further required to have an average first spike latency of less than 15 ms for the first stimulus and less than 20 ms for the last stimulus in the train to ensure the recorded neurons were not of the posteromedial nucleus. Finally, each stimulus-responsive neuron was assigned a primary whisker based on the whisker that evoked the earliest and largest response to the first stimulus in the train[32–35].

Previous work has shown that TRN axonal spikes can be recorded in extracellular recordings in the VPm and the mean waveform TPT width of these spikes is narrower than VPm spikes[9,39,42,43]. Thus, a final inclusion criterion for VPm neurons was that they must have a waveform width TPT of greater than 0.3 ms.

## Measures of neuronal activity

Change in ongoing firing rate due to the LED stimuli: It was necessary to quantify the properties of the neuronal circuit when L6CT neuron activity had reached a steady state from optogenetic manipulation. Thus, we omitted the early, more transient modulatory dynamics from our quantifications and defined the effect as the average firing rate across trials in the window from 250 ms (the end of the LED ramp) to 750 ms after the onset of the LED input. These firing rates were subtracted on a trial-by-trial basis from the pre-LED firing rate (spontaneous activity) in an equivalent 500 ms window.

Change in ongoing synchrony: To account for changes in firing rate between conditions, the synchrony strength was determined for pairs of neurons by computing the total numbers of spikes in a +/− 7.5 ms window in the resulting cross-correlogram and normalizing by the total number of spikes from each neuron[34,49]:

$$Strength = \frac{\sum ccg[-7.5:7.5ms]}{\sqrt{\frac{N^2_{neur1} + N^2_{neur2}}{2}}} \quad (1)$$

Sensory response: Sensory responses were quantified by looking at the average spike rate across trials in a 30 ms window post-sensory stimulus, subtracted on a trial-by-trial basis from the pre-sensory stimulus spike rate in an equivalent 30 ms window. The sensory response for each neuron was determined based on the stimulation of its primary whisker.

Burst spikes: VPm burst spikes are defined as two or more spikes occurring with an inter-spike interval of less than 4 ms, preceded by 100 ms or more of no spikes, as characteristic of T-type calcium channel burst spikes[33,35,75–77].

## Statistical analyses

The two-sided Wilcoxon signed rank test with a Bonferroni correction was used between pairs of LED conditions to test for statistically significant changes in various measures of the response of a respective population of neurons. The Spearman's rank correlation test was used to determine whether increasing LED intensity had a monotonic effect on changes in the ongoing firing rates of the respective population of neurons. When testing for changes in the ongoing activity of individual neurons, the spontaneous versus LED-evoked spike rates were compared using the two-sided Wilcoxon signed rank test. All tests were implemented using MATLAB built-in commands.

## LIF model

We constructed a simple circuit model that was representative of the key dynamics of the interactions between the L6CT neurons, the VPm, and the TRN. Rather than a largescale network model designed to represent the diversity of properties at the population level, the model we constructed was a reduced circuit model in the spirit of models previously used to describe thalamocortical dynamics[50], in which the average activity of neuronal populations is captured.

Model structure: The model consists of two main nodes, one representing the average VPm neuron response and the other representing the average TRN neuron response, both modeled as leak integrate-and-fire (LIF) neurons. Each node receives noisy excitatory inputs for spontaneous spiking. These inputs are modeled as homogeneous Poisson spike trains, each with its own firing rate. Both nodes also integrate the same 35 excitatory inputs that represent L6CT neuron spiking, modeled as homogenous Poisson spike trains. Finally, the TRN node forms an inhibitory synapse onto the VPm node.

Controlling L6CT synchrony: To control the synchrony of the L6CT inputs, the spike times for each of the L6CT spike trains are generated by drawing spikes from two homogeneous Poisson spike trains: one shared and one individual spike train, each with equivalent firing rates. For each synchrony condition, we selected the shared probability, a value between 0 and 1, that determines the likelihood that the spikes for each L6CT input would come from the shared spike train versus the individual spike train. The higher the shared probability, the more likely that the 35 L6CT inputs will have coincident spike times, increasing their synchrony, while maintaining equivalent firing rates.

Evolution of model dynamics: Each LIF node's membrane potential, $V$, evolves according to the membrane equation:

$$C_m \frac{dV}{dt} = -g_L(V - E_L) - \sum_j I^{syn}_j \quad (2)$$

where $C_m$ is the membrane capacitance, $g_L$ and $E_L$ are the leak conductance and reversal potential, respectively. $I^{syn}_j$ represents the synaptic currents received by each node from the $j^{th}$ source. Each source is modeled as one synapse according to the equation:

$$I^{syn}_j = w_j(V - E_j)g_j \quad (3)$$

where $w_j$, $E_j$, and $P_j$ are the weight, reversal potential, and conductance of the $j^{th}$ synapse. The synaptic conductance is incremented by $g_{jmax}$ after each presynaptic spike and decays according to the equation:

$$g_j = g_j - g_j \frac{dt}{\tau_{gj}} \quad (4)$$

where $\tau_{gj}$ is the respective synapses' decay constant. To account for synaptic transmission delays, a 1 ms delay is implemented between the time in which a pre-synaptic source fires a spike and the time in which this spike can begin to evolve the current in the post-synaptic neuron.

Finally, we implemented a 2 ms refractory period, during which the evolving currents for the node in a refractory period could not affect the neuron's membrane potential.

All model parameters were initially based on previously published parameters[78] and experimental observations[5]. The final parameters (Table 1) were then further tuned to better match the experimental neuronal response to the lowest LED input intensity, then allowing the response to all other LED inputs to emerge.

**Table 1 | Parameters used in the LIF model**

| Variable | Value used in model of VPm neuron | Value used in model of TRN neuron |
|---|---|---|
| $C_m$ [a] | 0.00029 μF | 0.00014 μF |
| $g_L$ | 0.029 μS | 0.007 μS |
| $E_L$ | −70 mV | −80 mV |
| $w_j$ | $w_{TRN}$ = 0.075<br>$w_{CT}$ = 1<br>$w_{Spont.Inp.}$ = 0.0129 | $w_{VPm}$ = 0<br>$w_{CT}$ = 1<br>$w_{Spont.Inp.}$ = 0.0057 |
| $E_j$ | $E_{TRN}$ = −80mV<br>$E_{CT}$ = 0mV<br>$E_{Spont.Inp.}$ = 0mV | $E_{VPm}$ = 0mV<br>$E_{CT}$ = 0mV<br>$E_{Spont.Inp.}$ = 0mV |
| $\tau_{P,j}$ | $\tau_{P,TRN}$ = 0.01s<br>$\tau_{g,CT}$ = 0.1s<br>$\tau_{g,Spont.Inp.}$ = 0.01s | $\tau_{P,TRN}$ = 0.01s<br>$\tau_{g,CT}$ = 0.1s<br>$\tau_{g,Spont.Inp.}$ = 0.01s |
| $g_{jmax}$ | $g_{TRN,max}$ = 1μS<br>$g_{CT,max}$ = 0.003μS<br>$g_{Spont.Inp.,max}$ = 1μS | $g_{VPm,max}$ = 1μS<br>$g_{CT,max}$ = 0.003μS<br>$g_{Spont.Inp.,max}$ = 1μS |
| $V_{threshold}$ | −54mV | −54mV |
| $V_{reset}$ | −80mV | −80mV |

[a]All italicized variables are defined in Eqs. 2–4

## Reporting summary

Further information on research design is available in the Nature Portfolio Reporting Summary linked to this article.

## Data availability

The processed data generated in this study have been deposited in the Zenodo database and can be accessed here: https://doi.org/10.5281/zenodo.8156087. Source data are provided with this manuscript. The raw data can be made available upon request. Source data are provided with this paper.

## Code availability

The code generated for all analyses in this study has been deposited in the Zenodo database and can be accessed here: https://doi.org/10.5281/zenodo.8156087.

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

## Acknowledgements
We thank Simone Russo and Kayla Peelman for their helpful comments on the manuscript. We thank Barry Connors' laboratory at Brown University for providing us with the NTSR1-cre mouse line.

## Author contributions
E.D. and G.B.S. conceptualized the project. E.D., A.P., and G.B.S designed the study. E.D. carried out all experiments. E.D. and V.C. performed the analysis. E.D., A.P., and G.B.S. wrote the manuscript with comments from V.C. and N.C.W. A.P., N.C.W., and G.B.S. supervised the project. E.D. and G.B.S acquired funding.

## Funding
This work was supported by NIH National Institute of Neurological Disorders and Stroke BRAIN Grant (NINDS) R01NS104928, NINDS BRAIN RF1NS128896, NINDS R21 NS112783. EDD is supported by a National Science Foundation Graduate Research Fellowship and the Howard Hughes Medical Institute through the James H. Gilliam Fellowships for Advanced Study program.

## Competing interests
The authors declare no competing interests.
