## [Peer Review File · Nature Communications]

Dynamic corticothalamic modulation of the somatosensory thalamocortical circuit during wakefulnessREVIEWER COMMENTS

Reviewer #1 (Remarks to the Author):

Dimwamwa et al. have manipulated layer 6 corticothalamic (L6 CT) activity in the whisker primary somatosensory cortex (S1) using a cell-type specific mouse driver line to study the functional impact on thalamocortical circuit excitability during wakefulness. Using a combination of optogenetics and multi-site in vivo electrophysiology, the authors report that in awake mice, activation of L6 CT cells using the *Ntsr1-Cre* driver line results in both suppression and enhancement of spiking activity in the ventral-posterior-medial nucleus of the thalamus (VPM) depending on the stimulation intensity/firing frequency of CT cells. The authors then demonstrate that inhibitory thalamic reticular nucleus neurons (TRN) do not show similar bi-directional modulation and, importantly, less spiking modulation than VPM cells during different CT activity patterns. These observations are largely consistent with previous in vitro work describing the role of short-term synaptic dynamics in switching thalamic excitability from suppression to enhancement (Crandall et al., 2015, *Neuron*). However, in addition to frequency, the authors also elegantly demonstrate, using both physiological recordings and modeling, that synchrony of the CT population plays a critical role in contributing to the modulatory effect on VPM activity. Despite the frequency-dependent bidirectional change in VPM spiking activity, the authors find that the sensory-evoked responses of VPM cells are not, but instead, consistently amplified with CT activity. Given L6 stimulation influences both thalamus and intracortical circuits, the authors next probed the impact of CT stimulation on ongoing activity and sensory responses of excitatory (RS) and inhibitory (FS) cortical neurons. They report that L6 CT activity has a net suppressive effect on the ongoing baseline activity of both RS and FS neurons across cortical S1. This observation is consistent with the idea that L6 is an intracortical gain control (Olsen et al., 2012, *Nature*). However, the net effect on sensory responses was more complex, with RS responses unchanged and FS responses dependent on the light intensity.

This study is technically impressive, with some interesting results. The observation that optical stimulation of L6 CT cells can influence ongoing thalamic and cortical spiking is not novel, but the use of awake, head-fixed mice is powerful, helping illustrate that such influences are intact during wakefulness. The authors also nicely highlight in their manuscript the role of CT synchrony in modulating thalamic excitability, which has received less attention but is important to the field. Finally, combining optical and sensory stimulation reveals how L6 CT cells can influence the sensory-evoked responses of different cells within the thalamocortical circuit. Technically, the authors combined in vivo multi-site electrophysiological recordings and optical techniques in a nicely integrated way. The paper is well-written and illustrated.

My questions and concerns are listed in roughly decreasing order of importance:

1. The observation that low-frequency CT activity (low-light intensities) modulates spontaneous VPM spiking activity but not sensory-evoked responses was a bit surprising. Why did the authors only choose to present the sensory stimulus 750 ms after the onset of the light stimulation? As stated, during the hold period of the LED input, the firing rate of the VPM cells increases throughout (although still suppressed). From the data presented, the transient and early hold period of the low-frequency CT condition is where the maximal suppression occurs, not at the end. So, a simple prediction would be that sensory responses in VPM neurons would be suppressed during this early transition phase and less during the late steady-state phase. Such a prediction would follow the underlying conductances previously shown in vitro.
2. Related to point 1 above, why not test the impact of CT synchrony on the VPM sensory responses?
3. It would be interesting to know if the authors observed any correlated firing changes among recorded VPM cells during LED modulation with and without whisker stimulation.

4. The authors state that 60% of the recorded VPM neurons are suppressed, and 25% are enhanced at 8 mW. Are these responses related to the functional topography of corticothalamic feedback stimulation and the VPM cell recorded?

5. Did all 27 TRN cells show the same response profile with LED modulation (Figure 3)?

6. Discussion: The authors demonstrate that L6 CT synchrony plays an important role in the modulation of VPM ongoing activity. It may be worth discussing and considering how L6 CT cells with a wide range of axon conduction velocities might contribute to this synchrony-dependent modulation of VPM.

7. Results Line 703-705 and Discussion Line 832-834: The authors should be more explicit about what they mean by "the suppression of the ongoing activity of both RS and FS S1 subpopulations likely arises predominantly from direct intracortical L6 CT projections,.". Are they saying that the suppression is caused directly by CT cells, which are excitatory? They probably mean the CT recruitment of inhibitory translaminar cells.

Reviewer #2 (Remarks to the Author):

Dimwamwa et al use in their manuscript titled "Dynamic corticothalamic modulation of the somatosensory thalamocortical circuit during wakefulness" optogenetic stimulation of cortical layer 6 neurons in barrel cortex and multi-site electrophysiological recordings in primary somatosensory thalamus to study the thalamocortical loop. Additionally, they build a model of this circuit to interpret their results. The manuscript is well written, carefully prepared, and the figures are beautiful. However, there are several major concern which might affect the interpretation of all findings.

Major concern:

The authors use optogenetic activation of layer 6 neurons in barrel cortex in Ntsr1-Cre mice. Barrel cortex in mice has a thickness of about 1200 μm . Layer 4 starts in a depth of about 500 μm below dura mater and layer 6 in about 1000 μm depth. The dendrites of L6CT neurons reach up to layer 4 (see Fig 1a). The authors place the light source for optogenetic stimulation on top of layer 1. To activate channelrhodopsin-2 expressed in L6CT neurons the light stimulus has to pass at least 500 μm through layer 1-3 to reach the dendrites of L6CT neurons in layer 4. However, light will also activate – without any optogenetic construct expressed – neurons. My concern is that dendrites and axons in layer 1 will get activated together with L6CT dendrites expressing channelrhodopsin-2 in layer 4. If this is the case the activation is completely uncontrolled. The authors must do the following control experiment: Use an Ntsr1-Cre mouse as in all other experiments but without injection of the AAV delivering the gene of the optogenetic tool and repeat the experiment with the same laser intensity. My concern is that the authors will find the same results.

An additional concern is that it was previously shown that the thalamocortical loop shows a strong dependency on behavioral state. This should be taken into account. Currently, only sensory stimuli are presented. An alert mouse will show different spike rates than a sleepy mouse.

The proposed model is – on purpose and justifiably – kept simple. However, it should be taken into account that the input onto TC neurons from L6CT neurons are not driving but causing NMDA plateau potentials in TC distal dendrites and thereby just changing the gain.

Dimwamwa et al., Dynamic corticothalamic modulation of the somatosensory thalamocortical circuit during wakefulness

Response to Reviewer Comments

We thank the reviewers for their efforts as well as their excellent and insightful comments on this manuscript. We have significantly modified our manuscript based on these comments and believe that the revised manuscript is substantially improved as a result. In addition to minor edits, we have conducted additional experiments to control for effects of the LED alone. We have explored the relationship between the effects of L6CT activation on ongoing vs sensory-driven VPM spiking and more thoroughly assessed L6CT and VPM synchrony in our manipulations as well as providing more insight into the activity of individual VPM and TRN neurons in our manipulations. We have expanded our analyses to account for the impact of the animals' behavioral state on our observations, and we have expanded the presentation and discussion of our results in the context of the array of mechanisms that could underlie our observations. Lastly, we have also updated the color scheme used throughout all figures to be inclusive of individuals with red-green color blindness.

We have retained the reviewers' original comments below in black and numbered the comments from reviewer 2. Our responses are in blue text, and any changes to the manuscript are in red text, along with line numbers where the changes appear in the revised manuscript.

Reviewer #1 (Remarks to the Author):

Dimwamwa et al. have manipulated layer 6 corticothalamic (L6 CT) activity in the whisker primary somatosensory cortex (S1) using a cell-type specific mouse driver line to study the functional impact on thalamocortical circuit excitability during wakefulness. Using a combination of optogenetics and multi-site in vivo electrophysiology, the authors report that in awake mice, activation of L6 CT cells using the Ntsr1-Cre driver line results in both suppression and enhancement of spiking activity in the ventral-posterior-medial nucleus of the thalamus (VPM) depending on the stimulation intensity/firing frequency of CT cells. The authors then demonstrate that inhibitory thalamic reticular nucleus neurons (TRN) do not show similar bi-directional modulation and, importantly, less spiking modulation than VPM cells during different CT activity patterns. These observations are largely consistent with previous in vitro work describing the role of short-term synaptic dynamics in switching thalamic excitability from suppression to enhancement (Crandall et al., 2015, Neuron). However, in addition to frequency, the authors also elegantly demonstrate, using both physiological recordings and modeling, that synchrony of the CT population plays a critical role in contributing to the modulatory effect on VPM activity. Despite the frequency-dependent bidirectional change in VPM spiking activity, the authors find that the sensory-evoked responses of VPM cells are not, but instead, consistently amplified with CT activity. Given L6 stimulation influences both thalamus and intracortical circuits, the authors next probed the impact of CT stimulation on ongoing activity and sensory responses of excitatory (RS) and inhibitory (FS) cortical neurons. They report that L6 CT activity has a net suppressive effect on the ongoing baseline activity of both RS and FS

neurons across cortical S1. This observation is consistent with the idea that L6 is an intracortical gain control (Olsen et al., 2012, Nature). However, the net effect on sensory responses was more complex, with RS responses unchanged and FS responses dependent on the light intensity.

This study is technically impressive, with some interesting results. The observation that optical stimulation of L6 CT cells can influence ongoing thalamic and cortical spiking is not novel, but the use of awake, head-fixed mice is powerful, helping illustrate that such influences are intact during wakefulness. The authors also nicely highlight in their manuscript the role of CT synchrony in modulating thalamic excitability, which has received less attention but is important to the field. Finally, combining optical and sensory stimulation reveals how L6 CT cells can influence the sensory-evoked responses of different cells within the thalamocortical circuit. Technically, the authors combined in vivo multi-site electrophysiological recordings and optical techniques in a nicely integrated way. The paper is well-written and illustrated.

We sincerely thank the reviewer for their encouragement and constructive feedback.

My questions and concerns are listed in roughly decreasing order of importance:

R1.1. The observation that low-frequency CT activity (low-light intensities) modulates spontaneous VPM spiking activity, but not sensory-evoked responses was a bit surprising. Why did the authors only choose to present the sensory stimulus 750 ms after the onset of the light stimulation? As stated, during the hold period of the LED input, the firing rate of the VPM cells increases throughout (although still suppressed). From the data presented, the transient and early hold period of the low-frequency CT condition is where the maximal suppression occurs, not at the end. So, a simple prediction would be that sensory responses in VPM neurons would be suppressed during this early transition phase and less during the late steady-state phase. Such a prediction would follow the underlying conductances previously shown in vitro.

The reviewer brings up a very important point concerning the experimental design and we thank them for this. It is increasingly being demonstrated that L6CT neurons modulate the thalamocortical circuit in contexts that occur on a relatively long timescale (e.g. during whisking, locomotion, head rotation, licking, etc.). So this study was designed to evaluate the effect of L6CT feedback on sensory responses throughout the thalamocortical circuit on timescales relevant to these contexts, while L6CT activity is in a steady-state. This is why we presented the sensory stimulus 750 ms after the initiation of the LED, avoiding the initial transients.

However, the reviewer brings up an interesting and reasonable prediction concerning the surprising nature of our results that the sensory response of VPM neurons is not suppressed under conditions where the ongoing activity is suppressed. We find that the effect of L6CT activation on the ongoing activity of VPM neurons does not trivially predict the effect of the activation on VPM neurons' sensory response. Specifically, in the figure below which is now included in the revised manuscript as Supplemental Figure S11, we

analyzed only neurons at any LED intensity whose ongoing activity is suppressed with L6CT activation. When plotting the change in ongoing firing rate against the difference in the sensory response between the no LED and LED-on conditions, we observe a Spearman's correlation coefficient of -0.6 . This indicates that the most suppressed neurons rather tend towards an increase in their sensory response. Importantly, this increase is not due to burst spikes (Figure S12).

Figure S11 | The effect of the LED modulation on the ongoing firing rate of VPM neurons does not trivially predict the effect on the neurons' sensory response. A simple prediction would be that for neurons whose ongoing firing rate is suppressed by the LED inputs, their sensory response would likewise be reduced. Thus, we present the change in ongoing firing rate of individual VPM neurons due to LED activation of L6CT neurons against the difference in sensory response between no LED and LED-on conditions. Note that we only include neurons' activity under conditions where the ongoing firing rate is significantly suppressed by LED inputs using the two-sided Wilcoxon signed rank test. Thus, the set of datapoints represent the response of neurons across all tested LED intensities. Under this set, there is a moderate, negative Spearman's correlation, suggesting a trend where the neurons whose ongoing activity is most suppressed by an LED input rather have a greater sensory response compared to no LED conditions, contrary to the simple prediction. Note that this trend holds even if we only include data under the 4 mW/mm² intensity condition (not shown).

Notably, the simple prediction is supported by the measured effect of L6CT activation on the underlying conductances and response to medial lemniscus stimulation previously measured in VPM *in vitro* by Crandall et al., 2015. But, in addition to differences in experimental preparation, measurements in the Crandall study were made in response to brief pulses of optogenetic activation, which likely would modulate the cortico-thalamocortical circuit in quite a different manner compared to the longer timescale manipulations in our study. Thus, it is not immediately obvious what the previous study would predict in the different stimulus conditions utilized in our present study.

The main manuscript text has now been revised to include the figure and the following revision to the main manuscript text to refer to this analysis and discussion:

“To assess the effect of L6CT modulation on the transfer of sensory inputs through VPM in a slice preparation, Crandall et al. co-activated L6CT neurons and medial lemniscus (ML) axons. They found a clear relationship between the effect of L6CT activation on the response of VPM neurons to ML stimulation and the biphasic modulatory effect on spontaneous VPM spiking (Crandall et al., 2015). From such a study, it is reasonable to expect that VPM sensory responses would be suppressed in conditions where VPM ongoing activity is suppressed, at 4 and 8 mW/mm². Yet, we rather observe that neurons with increasingly suppressed ongoing activity tend towards a more enhanced sensory response compared to the no LED condition (Figure S11). While one might expect an increase in T-type calcium channel-mediated burst spikes in conditions where the VPM ongoing activity is suppressed (Borden et al., 2022; Kirchgessner et al., 2020), we do not observe any significant changes in stimulus-evoked VPM bursting upon L6CT activations (Figure S12). Thus, the observed bidirectional changes in VPM ongoing activity do not trivially predict the VPM sensory response magnitudes, as sensory responses are consistently amplified by L6CT activation.” lines 705-719

R1.2. Related to point 1 above, why not test the impact of CT synchrony on the VPM sensory responses?

We thank the reviewer for highlighting this point, as the contribution of the synchrony of the L6CT neurons towards its function is an important takeaway from this work. We would like to clarify that the results we previously presented concerning the effect of L6CT neurons on sensory responses throughout the TC circuit are during the ramped step LED manipulations. Under these conditions, L6CT synchrony *is* varying, but so is L6CT firing rate (please see Figure 4), precluding a selective measure of the effects of L6CT synchrony on the VPM stimulus-evoked response.

It is natural to wonder about the selective effect of L6CT synchrony in accordance with our manipulations that varied synchrony while maintaining the firing rate constant. Thus, in a subset of experiments, we presented sensory stimuli during one of three conditions: no LED, the previously presented LED step at 16 mW/mm² intensity which we refer to here as the 0 Hz input, and the 500 Hz noise input with a mean intensity of 16 mW/mm². In this subset of data, presented in the figure below and now included in the revised manuscript as Supplemental Figure S10, the trend suggests that the VPM population sensory response is slightly enhanced across all LED conditions, however, there is no statistical significance. More relevant, the sensory response during the 500 Hz input is also unchanged compared to the sensory response during the 0 Hz input.

The main manuscript text has now been revised to include the figure and the following revision to the main manuscript text to refer to this analysis:

“Interestingly, the VPM population sensory response is not further affected when we present the sensory stimulus during the 500 Hz noise LED input, as compared to the 0

Hz LED step of equal mean. This suggests that a selective increase in the synchrony of the L6CT inputs does not further impact the sensory responses (Figure S10), in contrast to what is observed for ongoing VPM activity (Figure 5).” lines 699-703

Figure S10 | Increased L6CT synchrony does not differentially affect VPm sensory responses. **a.** Grand PSTHs (mean +/- sem across neurons) of the VPm population sensory response with no LED, the 0 Hz LED step, and the 500 Hz frequency modulated LED input, which selectively increases L6CT synchrony and not firing rate when compared to the 0 Hz LED step. Note that this population is a subset of the population presented in Figure 7. The right panel is an expanded view of the same population’s sensory response. Note the change in the vertical scale in the right and left views. **b.** While there is an increasing trend that matches the results presented in Figure 7, the various LED inputs have no significant effect on the baseline-subtracted, stimulus-evoked response of this subset of VPm neurons contributing to the PSTHs in **a** (+/- sem). Most relevant, there is no further boost in sensory response between the 0 and 500 Hz conditions (pairwise comparisons are to the No LED or 0 Hz condition using the two-sided Wilcoxon signed rank test with a Bonferroni correction).

R1.3. It would be interesting to know if the authors observed any correlated firing changes among recorded VPM cells during LED modulation with and without whisker stimulation.

This is a good point. This information was not provided in the original version of the manuscript. The analysis will be explained in detail below, but to summarize, we observe no change in the synchrony of the ongoing activity of the VPm population across all LED-on conditions and an increase in the stimulus-evoked synchrony strength across all LED-on conditions.

1. For the effect of the LED modulation on the synchrony of ongoing VPm population activity, we analyzed the synchrony of the ongoing activity of simultaneously recorded VPm neurons during varying levels of LED activation of L6CT neurons. Because the synchrony calculation involves normalizing by the mean firing rate, the resultant synchrony metric reflects the synchrony change beyond what would be predicted by the change in firing rate alone. We found that for low levels of LED, there was a slight decrease in VPm synchrony and a slight increase for high levels of LED, but none of these changes were statistically significant. This new analysis is presented below and is included as supplemental Figure S6:

LED modulation of the VPM population ongoing synchrony (166 neuron pairs)

Figure S6 | The effect of optogenetic activation of L6CT neurons on the synchrony of ongoing VPM activity is small and nonsignificant. The mean +/- sem change in the VPM population ongoing synchrony strength to ramped LED steps of increasing intensity (statistical comparisons are between the No LED condition and the respective LED-on conditions using the two-sided Wilcoxon signed rank test with a Bonferroni correction).

This new analysis is described in the revised manuscript as follows: “Intriguingly, while we observe a bidirectional trend of the effect of the LED manipulation of L6CT neurons on the ongoing synchrony of the VPM population that parallels the trend of the VPM firing rate presented in Figure 2, the changes are small and nonsignificant (Figure S6).” lines 563-566

- For the effect of the LED modulation on the synchrony of VPM population sensory responses, the analysis reveals that L6CT activation significantly synchronizes the stimulus-evoked VPM response, increasing steadily for increasing LED intensities. We now include Supplemental Figure S9 in the revised manuscript, also presented below:

LED modulation of the VPM population sensory response synchrony (166 neuron pairs)

Figure S9 | Increasing optogenetic activation of L6CT neurons increases the synchrony of VPM sensory responses. The mean +/- sem VPM population sensory response synchrony strength to ramped LED steps of increasing intensity (statistical comparisons are between the No LED and the respective LED-on conditions using the two-sided Wilcoxon signed rank test with a Bonferroni correction).

Correspondingly, we have revised the manuscript to reflect this finding: “We also observe a corresponding increase in the sensory response synchrony strength of the VPm population response across all LED intensities (Figure S9).” lines 695-697

R1.4. The authors state that 60% of the recorded VPM neurons are suppressed, and 25% are enhanced at 8 mW. Are these responses related to the functional topography of corticothalamic feedback stimulation and the VPm cell recorded?

This is a good and important question. Indeed, L6CT feedback to its thalamic targets is topographically aligned but the topographically diffuse TRN projections result in interesting surround suppression effects in first-order thalamic nuclei (Born 2021; Temereanca 2004).

As compared to the aforementioned studies that conducted focal activations of L6CT neurons to relate the functional topography of L6CT feedback and its effects on first-order thalamic nuclei, the spatially global nature of our manipulations results in a more complex relation between the functional topography of L6CT neurons and the effects in VPm. Specifically, our experimental design consisted of activating L6CT neurons with an optical fiber that is 400 μm in diameter and positioned at the cortical surface. 400 μm spans L6CT neurons in 2-4 functional, barrel columns. Further, the light must travel through hundreds of microns of tissue before reaching L6CT axons and cell bodies, which would further widen the spread of light, thereby activating L6CT neurons across even more functional columns. In short, we designed our manipulation to be a broad activation of L6CT neurons throughout the barrel cortex, representative of contexts such as locomotion (Augustinaite 2020), licking (Clayton 2021), and head rotation (Velez-Fort 2018), which would likely activate L6CT neurons throughout the primary somatosensory cortex in a global manner.

To explore the relationship between the functional topography of L6CT neurons and the effect of our manipulations in VPm, we conducted a new set of analyses. Specifically, we provide below activity profiles from neurons in one of two recordings. In the first recording, the simultaneously recorded VPm neurons have the same primary whisker. Therefore, these neurons receive inputs from L6CT neurons of the same functional column. Yet, the LED response is variable across these neurons, with one enhanced across all LED intensities, one suppressed, two unmodulated, and one bidirectionally modulated neuron.

The second example shows simultaneously recorded VPm neurons with two different primary whiskers, thus receiving inputs from L6CT neurons of differing functional columns. Yet, two of the three neurons are bidirectionally modulated. Importantly, those two neurons have differing primary whiskers.

These examples suggest that the heterogeneity of effects in VPm neurons' ongoing activity is not trivially related to the functional topography of L6CT neurons.

Figure S11 | The effect of the LED modulation on the ongoing firing rate of VPM neurons does not trivially predict the effect on the neurons' sensory response. A simple prediction would be that for neurons whose ongoing firing rate is suppressed by the LED inputs, their sensory response would likewise be reduced. Thus, we present the change in ongoing firing rate of individual VPM neurons against the difference in sensory response between no LED and LED-on conditions. Note that we only include neurons' activity under conditions where the ongoing firing rate is significantly suppressed by LED inputs using the two-sided Wilcoxon signed rank test. Thus, the set of datapoints represent the response of neurons across all tested LED intensities. Under this set, there is a moderate, negative Spearman's correlation, suggesting a trend where the neurons whose ongoing activity is most suppressed by an LED input rather have a greater sensory response compared to no LED conditions, contrary to the simple prediction. Note that this trend holds even if we only include data under the 4 mW/mm² intensity condition (not shown).

The main manuscript text has now been revised to include the figure and the following revision to the main manuscript text to refer to this analysis:

“When counting individual neurons, the ongoing firing rate of 60% of recorded neurons (45 of 75) is suppressed at 8 mW/mm², whereas 60% (45 of 75) of the same set of recorded neurons are enhanced at 30 mW/mm² (Figure 2g). This heterogeneity of effects can be observed in simultaneously recorded VPM neurons with preferential responsiveness to the same whisker. Further, bidirectional modulation can be observed in simultaneously recorded neurons with preferential responsiveness to differing whiskers (Figure S4). Thus, likely due to the broad spatial nature of our LED input, the population heterogeneity is not trivially explained by the functional topography of L6CT axonal projections.” lines 502-507

R1.5. Did all 27 TRN cells show the same response profile with LED modulation (Figure 3)?

This is a good question. We agree that it is worth providing insight into the response of individual TRN neurons and have included this as Supplemental Figure S5, included below. As demonstrated in this figure, most TRN neurons showed the same response

profile as the example neuron shown in Figure 3, with enhanced spiking to 8 mW/mm² activation and relatively little to no further increase at higher intensities. However, the spiking of a few neurons was monotonically enhanced, the spiking of one neuron was maximal at 8 mW/mm², and another neuron at 16 mW/mm². Importantly, in none of the neurons was the spiking suppressed by the LED activation of L6CT neurons.

LED modulation of individual TRN neurons (27 neurons)

Figure S5 | The effect of the LED modulation of L6CT neurons on individual TRN neurons. The mean change in ongoing firing rate of all recorded TRN neurons illustrates relatively homogeneous modulatory effects across the population.

The manuscript text has been revised to refer to this analysis as follows:

“The spiking of the example TRN neuron presented in Figures 3c and 3d is strongly increased by the LED at both 8 mW/mm² and 30 mW/mm² (Figure 3e). **This strong effect of the LED modulation across all LED intensities largely holds for all recorded TRN neurons (Figure S5).**” lines 531-533

R1.6. Discussion: The authors demonstrate that L6 CT synchrony plays an important role in the modulation of VPM ongoing activity. It may be worth discussing and considering how L6 CT cells with a wide range of axon conduction velocities might contribute to this synchrony-dependent modulation of VPM.

We thank the reviewer for bringing up this important point. Indeed, it is well documented that L6CT neurons have a very wide range of axon conduction velocities, which would cloud the interpretation of our results implicating synchronous L6CT spiking. Specifically, our synchrony measure concerns L6CT spike times, yet variability in conduction velocity would negatively impact the summation of the L6CT-mediated conductances in the sub-

cortical structures, even in the case of perfectly coincident L6CT spiking. We offer the following discussion.

First, the NTSR1-cre mouse line enables activation of a sub-class of L6CT neurons with relatively fast conduction latencies in the range of 1-7 ms (reviewed in Briggs & Usrey 2008). This relatively narrow range of latencies would correspond to a fairly small variability in timing (even if uniform), likely not dramatically affecting our measure of synchrony that considers coincident spiking within a 15 ms time window (± 7.5 ms).

However, given the overall wide range of conduction velocities of all L6CT neurons, the full impact of the timing of the synaptic inputs to VPM and TRN in our experimental design is unknown. This variability in conduction velocity would likely increase the variability (and thus decrease the synchrony) of the synaptic inputs to VPM and TRN. While this could lessen the effect of the phenomenon we describe here, the window of opportunity for these descending inputs would still likely influence these sub-cortical structures in a synchrony-dependent manner. Nevertheless, we have noted this point in the revised manuscript.

We have updated the manuscript towards these important discussion points as follows:

“As suggested by the circuit model in this study, the L6CT synchrony-dependent, bidirectional modulation of VPM emerges, first, because of the strong inhibitory conductance from TRN inputs relative to the strength of the direct excitation from L6CT neurons. The second important feature is the circuit architecture that enables a monosynaptic excitatory conductance and an inherently delayed, disynaptic inhibitory conductance in the VPM. These elements are key to the “window of opportunity” classically described in the feedforward path for FO thalamic inputs onto layer 4 cortical RS neurons (Bruno, 2011; Isaacson & Scanziani, 2011; Pinto, 2003). The premise of this mechanism is that while the strength of individual thalamocortical synaptic inputs is modest, numerous inputs converge on cortical synaptic targets, making the cortical neurons highly sensitive to the synchrony of the ascending inputs (Bruno, 2011). **Indeed, L6CT inputs to VPM and TRN are numerous and highly convergent (Guillery & Sherman, 2002; Liu et al., 1995; Sherman & Koch, 1986b; Van Horn et al., 2000), contributing to a synchrony-dependent effect of this classically modulatory circuit. This would provide another example to support the possibility of a synchrony-driven window of opportunity as a canonical circuit computation in the brain.**

It is, however, important to note that there are several fundamentally different biological considerations between L6CT and thalamocortical inputs that would impact the window of opportunity in the feedback versus feed-forward circuits. First, L6CT neurons have a wide range of axon conduction latencies relative to that of thalamocortical projections. This would impact the synchronization of the post-synaptic conductances towards summation in the sub-cortical targets. It is worth noting that latencies for corticothalamic signal propagation in the range of 1-7 ms have been measured amongst the full range of

axon conduction latencies (reviewed in Briggs & Usrey, 2008). This would lead to a variability that is relatively small compared to the 15 ms (+/- 7.5 ms) window of opportunity that we consider here. Nonetheless, the full impact of the heterogeneity of L6CT conduction velocities in our study is currently unknown and should be pursued in future studies.” lines 862-873

R1.7. Results Line 703-705 and Discussion Line 832-834: The authors should be more explicit about what they mean by “the suppression of the ongoing activity of both RS and FS S1 subpopulations likely arises predominantly from direct intracortical L6 CT projections,”. Are they saying that the suppression is caused directly by CT cells, which are excitatory? They probably mean the CT recruitment of inhibitory translaminar cells.

We thank the reviewer for pointing out this lack of clarity and have updated the text to make it clear that we are referring to the latter: the L6CT recruitment of inhibitory cells. In the revised manuscript, the corresponding text now reads:

“This suppression likely arises in large part from L6CT direct intracortical projections to S1 inhibitory neurons, as previously characterized in the visual cortex (Bortone et al., 2014).” Line 742

“While optogenetic interventional approaches in the primary visual cortex have shown that the suppression of ongoing activity of both the RS and FS S1 subpopulations arises predominantly from direct intracortical L6CT projections to inhibitory translaminar cells (Bortone et al., 2014),...” lines 897-900

Reviewer #2 (Remarks to the Author):

Dimwamwa et al use in their manuscript titled “Dynamic corticothalamic modulation of the somatosensory thalamocortical circuit during wakefulness” optogenetic stimulation of cortical layer 6 neurons in barrel cortex and multi-site electrophysiological recordings in primary somatosensory thalamus to study the thalamocortical loop. Additionally, they build a model of this circuit to interpret their results. The manuscript is well written, carefully prepared, and the figures are beautiful. However, there are several major concerns which might affect the interpretation of all findings.

We sincerely thank the reviewer for their encouragement and for addressing very important points.

Major concern:

R2.1. The authors use optogenetic activation of layer 6 neurons in barrel cortex in Ntsr1-Cre mice. Barrel cortex in mice has a thickness of about 1200 um. Layer 4 starts in a

depth of about 500 μm below dura mater and layer 6 in about 1000 μm depth. The dendrites of L6CT neurons reach up to layer 4 (see Fig 1a). The authors place the light source for optogenetic stimulation on top of layer 1. To activate channelrhodopsin-2 expressed in L6CT neurons the light stimulus has to pass at least 500 μm through layer 1-3 to reach the dendrites of L6CT neurons in layer 4. However, light will also activate – without any optogenetic construct expressed – neurons. My concern is that dendrites and axons in layer 1 will get activated together with L6CT dendrites expressing channelrhodopsin-2 in layer 4. If this is the case the activation is completely uncontrolled. The authors must do the following control experiment: Use an Ntsr1-Cre mouse as in all other experiments but without injection of the AAV delivering the gene of the optogenetic tool and repeat the experiment with the same laser intensity. My concern is that the authors will find the same results.

This is an excellent point that was not addressed in the original manuscript. We presume that the reviewer is concerned about the effect of the light on the activation of the circuit that is not through the opsin itself, and not specifically an issue particular to the NTSR1-cre mouse. To control for confounds that could arise due to the LED alone, we recorded neuronal activity throughout the entire S1 cortical depth of an awake, wildtype mouse with no opsin expression. We compared spontaneous and stimulus-evoked spiking between trials with and without the LED at all intensities used in the manuscript. We found that the LED had minimal effects on the spiking activity, as demonstrated in the figure below which has now been included as Figure S1 in the manuscript.

We present the neuronal responses to 8 and 30 mW/mm^2 LED inputs from two recordings in the control animal. Importantly, we are including all possible neuronal clusters, including those that are not well-isolated, to determine if the LED illumination has any impact at all on spiking activity throughout the cortical depth. As evidenced by both the PSTHs and the population summary across cortical depth, the spontaneous spiking activity is not significantly affected by the LED at 8 and 30 mW/mm^2 , respectively (only one neuron at 8 mW/mm^2 and four at 30 mW/mm^2). Further, the sensory response was not affected by the LED for any of the neurons. Thus, the LED input alone cannot explain the observations in our manuscript.

The main manuscript text has now been revised to refer to this experiment and analysis as follows:

“All experiments were conducted in 11 male and 7 female NTSR1-cre (B6.FVB(Cg)-Tg(Ntsr1-cre)GN220Gsat/Mmcd, MMRRC) adult mice as well as 1 wildtype, C57BL/6J mouse, all aged 6 weeks to 6 months housed under a reversed light-dark cycle.” lines 123-125

“With the exception of the control experiments conducted in a wildtype mouse, all neuron clusters considered for analysis passed two criteria: a signal-to-noise ratio of the mean spike waveform greater than three and less than 2% of all spikes violating a 2 ms imposed refractory period.” lines 263-266

“Also note that the elicited changes in spiking activity were not due to the LED illumination alone, as verified in control experiments conducted in a wildtype mouse (Figure S1).” lines 460-462

Awake, WT mouse **Figure S1 | The LED input alone does not explain our observed effects of L6CT activation on the cortico-thalamocortical circuit.** **a.** The direct effect of the LED input alone on ongoing and stimulus-evoked cortical neuron activity is measured from in vivo, silicon probe recordings of neuronal spiking in S1 of an awake, head-fixed, wildtype (WT) mouse with no opsin expression. As in the experiments with opsin-expressing, NTSR1-cre mice, the optical fiber was positioned at the cortical surface and the effect of LED inputs of varying intensities on multi-unit neuron spiking was measured throughout the cortical depth. **b.** Grand PSTHs (mean \pm sem across neurons) of the cortical population in conditions with and without the LED at intensities of 8 (left) and 30 mW/mm^2 (right). The population's ongoing and sensory evoked activity are unchanged. **c.** The mean change in ongoing firing rate of individual cortical neurons due to LED inputs at 8 mW/mm^2 (left) and 30 mW/mm^2 (right), sorted by cortical depth and color-coded according to significance. Significance is determined by the Wilcoxon signed rank test on pre- and post-LED input firing rates across trials. Note that the x-axis range is that shown in Figure S13 for neurons in the NTSR1-cre mice. The spiking of the majority of neurons was unaffected by the LED inputs. **d.** Same as **c**, but for the sensory response. Note that the x-axis range is that shown in Figure S17 for neurons in the NTSR1-cre mice. The sensory response was unaffected by the LED inputs for all neurons.

R2. 2. An additional concern is that it was previously shown that the thalamocortical loop shows a strong dependency on behavioral state. This should be taken into account. Currently, only sensory stimuli are presented. An alert mouse will show different spike rates than a sleepy mouse.

The reviewer is bringing up an important point concerning behavioral state that was not sufficiently addressed in the original manuscript. Indeed it is well-documented that behavioral state and/or arousal influences overall spike rates throughout the thalamocortical circuit and thus could have influence over the measures presented here. In a wide range of studies, extrinsic metrics such as pupil dilation, whisking, and running, as well as internal measures of cortical state from the local field potential, have been shown to be correlated with behavioral state and overall arousal. We are unable to utilize our electrophysiological recordings to track cortical state here, as the optogenetic manipulation of the L6CT neurons strongly influences the LFP measure that we would otherwise use to classify cortical state (Guo, et al., 2014, Weiss et al., 2024). Similarly, the animals in our experiments are not on a treadmill or wheel, precluding measurements of the effects of locomotion. Finally, pupil measurements were not acquired in this study, unfortunately. However, we did track whisker movement, which is a correlate of cortical state (Poulet & Crochet, 2019). It is well known that whisking during sensory responses is correlated with state changes that can greatly impact spontaneous and sensory-driven spiking activity throughout the thalamocortical circuit, beyond what would just be expected through sensory reafference.

Note that the data originally presented in the manuscript selectively included trials in which there was no whisker movement. To investigate the issue raised by the reviewer, we re-analyzed data across epochs with both no whisker movement and whisker movement. Specifically, we conducted the four following analyses:

1. To determine whether any state changes associated with whisking have an effect on L6CT modulation of VPM ongoing activity, we measured the effect of our originally presented LED intensities on VPM ongoing activity in trials with whisker movement, as originally presented in Figure 2, and without whisker movement. This analysis is now included in the manuscript as Figure S3.

We found that although whisker movement is associated with an increase in the overall spontaneous firing rate of VPM neurons as shown by Urbain et al., 2015, the effect of the LED activation of the L6CT neurons is nearly identical for trials with and without whisker movement, as measured by the change in ongoing firing rate relative to no LED conditions.

The main manuscript text has now been revised to refer to this analysis as follows: **“The data presented here only includes trials with no whisker movement, a factor that is well known to be associated with changes in global brain state and modulation of VPM spiking (Urbain et al., 2015). Analysis of trials with whisker movement shows that the effect of the LED activation of L6CT neurons on the spontaneous activity of VPM**

neurons is nearly identical for trials with and without whisker movement (Figure S3).”
 lines 494-498

Figure S3 | The effect of the LED activation of L6CT neurons on VPM spontaneous activity is nearly identical for trials with and without whisker movement. a. The mean \pm sem change in ongoing activity of the VPM population at the various LED intensities in trials with (black) and without (gray) whisker movement. For this analysis, we only included experiments for which there was a minimum of five trials across all LED intensities for both whisker movement conditions. The gray data points thus reflect a subset of the neurons in Figure 2. Note that a more strict requirement on the minimum allowable trials does not change the nature of the results. The color indicates the p-value using the two-sided Wilcoxon signed rank test with a Bonferroni correction. **b.** Confusion matrix indicating the significance level of the change in ongoing firing rate of the VPM population observed in **a** when comparing between trials with and without whisker movement at each LED intensity.

2. To determine whether any state changes associated with whisker movement have an effect on L6CT modulation of VPM sensory responses, we measured the effect of our originally presented LED intensities on VPM stimulus-evoked firing rates in trials with whisker movement, as originally presented in Figure 7, and without whisker movement. This analysis is now included in the manuscript as Figure S8 and is included below.

We found that whisker movement was associated with a decrease in the L6CT-modulated sensory response in VPM across all LED-on conditions. Additionally, while increasing LED intensity increased the sensory response in trials *without* whisker movement (Figure 7), in trials *with* whisker movement, the LED had no impact on the sensory response compared to the baseline, no LED condition. Thus, whisker movement is associated with a reduction of the measured effect of L6CT activation on VPM sensory responses.

Figure S8 | The measured effect of L6CT activation on VPM sensory responses in trials with whisker movement is reduced compared to the effect in trials without whisker movement. **a.** The mean \pm sem sensory response of the VPM population at the various LED intensities in trials with (black) and without (gray) whisker movement. For this analysis, we only included experiments for which there was a minimum of five trials across all LED intensities for both whisker movement conditions. The gray data points thus reflect a subset of the neurons in Figure 7. Note that a more strict requirement on the minimum allowable trials does not change the nature of the results. **b.** Confusion matrix indicating the significance level of the sensory response of the VPM population observed in **a** when comparing between trials with and without whisker movement at each LED intensity. The color indicates the p-value using the two-sided Wilcoxon signed rank test with a Bonferroni correction. **c.** Confusion matrix indicating the significance level of the VPM population sensory response between no LED and LED-on conditions observed only in trials with whisker movement; the color indicates the p-value using the two-sided Wilcoxon signed rank test with a Bonferroni correction. No conditions are different from each other.

The main manuscript text has now been revised to refer to this analysis as follows: “These effects are measured in trials without whisker movement, as presented in Figure 7. The effects in trials with whisker movement are associated with a reduction of the overall magnitude of the VPM sensory response as well as a reduction in the measured effect of L6CT activation on the VPM sensory response (Figure S8).” lines 691-695

- To determine whether any state changes associated with whisker movement have an effect on L6CT modulation of S1 ongoing activity, we measured the effect of our originally presented LED intensities on S1 RS and FS ongoing activity in trials with whisker movement, as originally presented in Figure 8, and without whisker movement. This analysis is now included in the manuscript as Figure S15 and is included below.

We found that whisker movement is associated with a modest decrease in the measured effect of L6CT activation on RS ongoing activity but is not associated with any changes in the effect of L6CT activation on FS ongoing activity.

The main manuscript text has now been revised to refer to this analysis as follows: “Despite the well-documented effects of behavioral state changes reflected in whisker movement on overall cortical spiking (reviewed in Poulet & Crochet, 2019), whisker movement was associated with only a modest decrease in the measured effect of

L6CT activation on ongoing RS activity and was not associated with any change in the effect of L6CT activation on ongoing FS activity (Figure S15).” lines 756-760

Figure S15 | Compared to the effect in trials *without* whisker movement, while the measured effect of L6CT activation on ongoing S1 RS activity is modestly decreased in trials *with* whisker movement, the effect on ongoing S1 FS activity is not associated with any change. a. The mean \pm sem change in ongoing activity of the S1 RS (non-L6CT) population at the various LED intensities in trials with (black) and without (gray) whisker movement. For this analysis, we only included experiments for which there was a minimum of five trials across all LED intensities for both whisker movement conditions. The gray data points thus reflect a subset of the neurons in Figure 8. Note that a more strict requirement on the minimum allowable trials does not change the nature of the results. **b.** Confusion matrix indicating the significance level of the change in ongoing firing rate of the RS population observed in **a** when comparing between trials with and without whisker movement at each LED intensity. The color indicates the p-value using the two-sided Wilcoxon signed rank test with a Bonferroni correction. **c-d.** Same as **a-b**, but for the S1 FS population.

4. To determine whether any state changes associated with whisker movement have an effect on L6CT modulation of S1 sensory responses, we measured the effect of our originally presented LED intensities on RS and FS stimulus-evoked firing rates in trials

with whisker movement, as originally presented in Figure 8, and without whisker movement. This analysis is now included in the manuscript as Figure S16 and is included below.

We found that whisker movement was associated with a decrease in the L6CT-modulated sensory response across all LED intensities for both sub-populations. Similar to the results in trials *without* whisker movement (as presented in Figure 8), in trials *with* whisker movement, the LED has no impact on the RS sensory response compared to the no LED condition. For the FS sub-population, while the trends observed in trials *with* whisker movement hold true in trials *without* whisker movement, the enhancement of the sensory response at 4 mW/mm² is not significant in trials with whisker movement.

The main manuscript text has now been revised to refer to this analysis as follows:

“Whisker movement is associated with an overall decrease in RS sensory responses across all LED intensities. Further, pairwise comparisons of the RS population’s sensory response between no LED and LED-on conditions are also nonsignificant in trials with whisker movement (Figure S16).” Lines 768-771

“Despite an overall decrease in sensory response across all LED intensities in trials with whisker movement, the trend of an enhanced response at 4 mW/mm² and suppressed response at higher intensities is also present but nonsignificant in trials with whisker movement (Figure S16).” lines 783-786

Figure S16 | The overall impact of L6CT activation on S1 sensory responses is reduced in trials with whisker movement. **a.** The mean \pm sem sensory response of the S1 RS (non-L6CT) population at the various LED intensities in trials with (black) and without (gray) whisker movement. For this analysis, we only included experiments for which there was a minimum of five trials across all LED intensities for both whisker movement conditions. The gray data points thus reflect a subset of the neurons in Figure 8. Note that a more strict requirement on the minimum allowable trials does not change the nature of the results. **b.** Confusion matrix indicating the significance level of the sensory response of the RS population observed in **a** when comparing between trials with and without whisker movement at each LED intensity. The color indicates the p-value using the two-sided Wilcoxon signed rank test with a Bonferroni correction. **c.** Confusion matrix indicating the significance level of the RS population sensory response between no LED and LED-on conditions observed only in trials with whisker movement; the color indicates the p-value using the two-sided Wilcoxon signed rank test with a Bonferroni correction. No conditions are different from each other. **d-f.** Same as **a-c**, but for the S1 FS population.

R2.3. The proposed model is – on purpose and justifiably – kept simple. However, it should be taken into account that the input onto TC neurons from L6CT neurons are not driving but causing NMDA plateau potentials in TC distal dendrites and thereby just changing the gain.

As noted by the reviewer, it is well documented that L6CT neurons synapse onto distal dendrites of thalamic neurons and cause NMDA plateau potentials that are modulatory in nature as opposed to driving. As a result, individual L6CT inputs are weak and have longer decay constants as compared to AMPA-driven potentials such as those observed at the thalamocortical synapse. Despite the simplicity of our single-compartment model,

such aspects undoubtedly contribute to our experimental results that the model is intended to recapitulate. So we thank the reviewer for bringing up this important point and have adjusted the model parameters and results. Most importantly, we increased the time constants of L6CT synaptic inputs to be an order of magnitude longer than the other synapses in the network. Please see the updated figure below, for which the main message remains unchanged.

Figure 6 | Modeling the role of excitatory and inhibitory conductances in the L6CT synchrony-dependent bidirectional modulation of VPM ongoing activity. **a.** Representative VPM and TRN neurons were modeled as leaky, integrate and fire (LIF) neurons, each integrating common L6CT spike train inputs with controlled synchrony and firing rate. The VPM neuron integrated inhibitory input from the spiking of the TRN neuron. Spontaneous activity for each LIF neuron was driven by noisy, excitatory inputs. **b.** Single trial model simulations run by holding the L6CT neurons' firing rates fixed while varying the synchrony of the L6CT input result in a shift from suppression to enhancement of VPM spiking in a graded manner with increasing synchrony. Note that the spontaneous conductance has been scaled 5x for better visualization. **c.** The mean (\pm sem across 50 trials) change in VPM firing rate with increasing L6CT population synchrony, but constant L6CT neurons' firing rate (2.5 sp/s) replicates the experimentally observed, L6CT synchrony-dependent, bidirectional modulation of VPM. **d.** An expanded view of the single trial components and the underlying conductances in VPM highlights the key factors underlying L6CT synchrony-dependent, bidirectional modulation of VPM: (1) the relative strength of the inhibitory conductance compared to the excitatory conductance, and (2) the relative timing of the monosynaptic excitatory and disinaptic inhibitory conductances.

Additionally, the main manuscript text has now been revised to refer to these important biological considerations as follows:

“There are several well-documented factors that undoubtedly play a role in our experimental findings, such as the short-term dynamics of the involved synapses (Crandall et al., 2015) as well as the fact that L6CT inputs result in modulatory NMDA plateau potentials in the distal dendrites of FO thalamic neurons (Sherman & Guillery,

1998). We propose that the interplay of both the relative strength and timing of the excitatory and inhibitory conductances, as observed in our model, is a relatively simple mechanism that plays an additional key role in our experimental findings.” Lines 657-663

“It is, however, important to note that there are several fundamentally different biological considerations between L6CT and TC inputs that would impact the window of opportunity in the feedback versus feed-forward circuits. ...

In addition, it is well known that thalamocortical inputs produce fast conductance changes on the proximal dendrites of their postsynaptic targets via ionotropic receptors. However, L6CT neurons activate both ionotropic and metabotropic receptors, the latter of which produces prolonged NMDA plateau potentials in the distal dendrites of FO thalamic neurons (Sherman & Guillery, 1998). While the time constants of the L6CT synapses in our model were adjusted to reflect the longer timescale of NMDA plateau potentials (Table 1), the nature of our single-compartment model precludes capturing such biological complexity. It is nonetheless important to note that such factors undoubtedly contribute to our experimental results and would impact the window of opportunity.” Lines 862-882

Lastly, Table 1 has been updated with the appropriate parameters, as highlighted in red below.

Table 1. Parameters used in the LIF model		
Variable	Value used in model of VPm neuron	Value used in model of TRN neuron
C_m	0.00029 μF	0.00014 μF
g_L	0.029 μS	0.007 μS
E_L	-70 mV	-80 mV
w_j	$w_{TRN} = 0.075$ $w_{CT} = 1$ $w_{Spont.Inp.} = 0.0129$	$w_{VPm} = 0$ $w_{CT} = 1$ $w_{Spont.Inp.} = 0.0057$
E_j	$E_{TRN} = -80$ mV $E_{CT} = 0$ mV $E_{Spont.Inp.} = 0$ mV	$E_{VPm} = 0$ mV $E_{CT} = 0$ mV $E_{Spont.Inp.} = 0$ mV
$\tau_{P,j}$	$\tau_{P,TRN} = 0.01$ s $\tau_{g,CT} = 0.1$ s $\tau_{g,Spont.Inp.} = 0.01$ s	$\tau_{P,TRN} = 0.01$ s $\tau_{g,CT} = 0.1$ s $\tau_{g,Spont.Inp.} = 0.01$ s
g_{jmax}	$g_{TRN,max} = 1$ μS $g_{CT,max} = 0.003$ μS $g_{Spont.Inp.,max} = 1$ μS	$g_{VPm,max} = 1$ μS $g_{CT,max} = 0.003$ μS $g_{Spont.Inp.,max} = 1$ μS
$V_{threshold}$	-54 mV	-54 mV
V_{reset}	-80 mV	-80 mV

REVIEWERS' COMMENTS

Reviewer #1 (Remarks to the Author):

The authors have addressed all of my previous concerns. In addition, they have performed several additional analyses/experiments that strengthen this study. The revised manuscript is well-written, and I enjoyed reading it. Finally, I want to congratulate the authors.

Comment

Perhaps a better reference than the Sherman & Guillery 1998 review to consider when discussing NMDA plateau potentials in TC cells (as mentioned by R2) would be Augustinaite et al. 2014. I don't think the Sherman & Guillery 1998 review discusses NMDA spike/plateau potentials.

Augustinaite, S., Kuhn, B., Helm, P.J., and Heggelund, P. (2014). NMDA spike/plateau potentials in dendrites of thalamocortical neurons. *J Neurosci* 34, 10892-10905.

Reviewer #2 (Remarks to the Author):

The authors did a great job dressing my questions.

The only mistake I found was in Line 877 of the updated manuscript:

The authors cite Sherman & Guillery, 1998, for the NMDA plateau potentials which is not correct. I would recommend, for example, Augustinaite et al, (2014) NMDA Spike/Plateau Potentials in Dendrites of Thalamocortical Neurons. *J Neurosci*.

Dimwamwa et al., Dynamic corticothalamic modulation of the somatosensory thalamocortical circuit during wakefulness

Response to Reviewer Comments

We thank the reviewers for their efforts as well as their excellent and insightful comments on this manuscript. We have made the requested change in citation.

We have retained the reviewers' original comments below in black. Our response is in blue text.

Reviewer #1 (Remarks to the Author):

The authors have addressed all of my previous concerns. In addition, they have performed several additional analyses/experiments that strengthen this study. The revised manuscript is well-written, and I enjoyed reading it. Finally, I want to congratulate the authors.

Comment

Perhaps a better reference than the Sherman & Guillery 1998 review to consider when discussing NMDA plateau potentials in TC cells (as mentioned by R2) would be Augustinaite et al. 2014. I don't think the Sherman & Guillery 1998 review discusses NMDA spike/plateau potentials.

Augustinaite, S., Kuhn, B., Helm, P.J., and Heggelund, P. (2014). NMDA spike/plateau potentials in dendrites of thalamocortical neurons. *J Neurosci* 34, 10892-10905.

Reviewer #2 (Remarks to the Author):

The authors did a great job dressing my questions.

The only mistake I found was in Line 877 of the updated manuscript:

The authors cite Sherman & Guillery, 1998, for the NMDA plateau potentials which is not correct. I would recommend, for example, Augustinaite et al, (2014) NMDA Spike/Plateau Potentials in Dendrites of Thalamocortical Neurons. *J Neurosci*.

We sincerely thank the reviewers for their encouragement and constructive feedback. We have made the requested citation change.